# MoPFormer: Motion-Primitive Transformer for Wearable-Sensor Activity Recognition

**Hao Zhang[1]\*, Zhan Zhuang[1,2]\*, Xuehao Wang[3], Xiaodong Yang[4], Yu Zhang[1]†**
[1]Southern University of Science and Technology, [2]City University of Hong Kong
[3]Zhejiang University, [4]Institute of Computing Technology, Chinese Academy of Sciences
zhanghao@mail.sustech.edu.cn, zhazhuang3-c@my.cityu.edu.hk
xuehaowangfi@gmail.com, yangxiaodong@ict.ac.cn
yu.zhang.ust@gmail.com

## Abstract

Human Activity Recognition (HAR) with wearable sensors is challenged by limited interpretability, which significantly impacts cross-dataset generalization. To address this challenge, we propose Motion-Primitive Transformer (MoPFormer), a novel self-supervised framework that enhances interpretability by tokenizing inertial measurement unit signals into semantically meaningful motion primitives and leverages a Transformer architecture to learn rich temporal representations. MoPFormer comprises two stages. The first stage is to partition multi-channel sensor streams into short segments and quantize them into discrete "motion primitive" codewords, while the second stage enriches those tokenized sequences through a context-aware embedding module and then processes them with a Transformer encoder. The proposed MoPFormer can be pre-trained using a masked motion-modeling objective that reconstructs missing primitives, enabling it to develop robust representations across diverse sensor configurations. Experiments on six HAR benchmarks demonstrate that MoPFormer not only outperforms state-of-the-art methods but also successfully generalizes across multiple datasets. More importantly, the learned motion primitives significantly enhance both interpretability and cross-dataset performance by capturing fundamental movement patterns that remain consistent across similar activities, regardless of dataset origin.

## 1 Introduction

Human Activity Recognition (HAR) has a wide range of applications and can be achieved through various methods [42, 4]. While vision-based and environmental sensor-based methods have been extensively explored [23, 18], systems leveraging Inertial Measurement Units (IMUs) offer distinct advantages, including compact size, low power consumption, and minimal computational overhead for data acquisition and preprocessing. IMUs also inherently preserve privacy compared to visual alternatives [10, 31] and are robust to environmental factors like lighting variations and occlusions [25]. Such characteristics make IMUs a highly suitable sensing modality for capturing human motion across diverse real-world scenarios.

Despite these strengths, IMU-based HAR confronts a significant hurdle: interpretability. IMU data, comprising multi-channel time series of linear acceleration and angular velocity, is inherently less intuitive for human to interpret than video data, making it hard to understand what the model has learned [19]. Moreover, many human activities exhibit hierarchical structures, composed of more

---

\*Equal contribution.
†Corresponding author.

39th Conference on Neural Information Processing Systems (NeurIPS 2025).

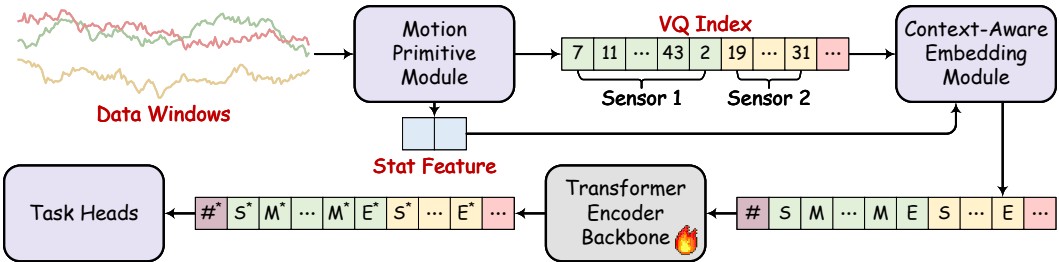

Figure 1: Architecture of the proposed MoPFormer model, showcasing the flow from raw data windows through tokenization, embedding, Transformer encoding, to task-specific heads.

fundamental activities or movements. However, there is a common absence of fine-grained labels for the sub-activities that constitute these complex, hierarchical activities. For example, an activity like "washing hands" can be deconstructed into sub-activities such as "turning on the tap", "applying soap", "scrubbing hands", and "rinsing hands". An HAR model might learn to identify "washing hands" by detecting the "turning on the tap" sub-activity, especially if it is a prominent signal. However, if the model encounters a different activity, such as "filling a water bottle", which also begins with "turning on the tap", it may struggle for HAR models to differentiate between the two activities. This is because the model might rely on the common initial sub-activity rather than grasping the complete, semantically meaningful sequence (e.g., the subsequent application of soap and scrubbing unique to handwashing) that defines the core of the activity. Each of these sub-activities might, in turn, be composed of even finer-grained movements. The black-box nature of contemporary HAR models often obscures whether these models are learning the underlying activity structure or merely exploiting superficial data correlations, thereby exacerbating the interpretability challenge [27, 3].

Another challenge in sensor-based HAR is data heterogeneity. Variations in sensor types, sampling rates, on-body placement (e.g., sensors worn on the wrist or the waist), subject characteristics (e.g., age, height, fitness level of the individual), and environmental conditions can introduce significant distributional shifts [32]. This heterogeneity impedes model generalization across different contexts. Consequently, developing strategies to address data heterogeneity is crucial for building robust HAR systems.

To address these challenges, we introduce **Mo**tion-**P**rimitive Trans**Former** (MoPFormer), a novel pre-training architecture for IMU-based HAR that significantly enhances model interpretability. Inspired by advancements in language modeling [20], MoPFormer conceptualizes IMU motion sequences as a series of "Motion Primitives" that serve as semantically meaningful, discrete building blocks of human activity. As illustrated in Fig. 1, MoPFormer first divides raw IMU sequences into short segments, analogous to "words" in natural language, creating an interpretable vocabulary of fundamental movements. Each segment integrates data from various sensor channels at concurrent time points, forming feature vectors that are then arranged sequentially into a two-dimensional feature matrix. This construction not only mirrors the arrangement of words in a sentence but also enables transparent analysis that which motion primitives contribute to activity recognition decisions. To further enrich semantic understanding, we embed metadata of each sensor using a context-aware embedding module. The resulting representation allows for direct examination of motion primitive similarities, frequencies, and their distribution across different activities, providing unprecedented insights into model behavior. This comprehensive representation is subsequently processed by a Transformer backbone [48], enabling MoPFormer to effectively handle heterogeneous channel configurations and learn robust, interpretable features for diverse downstream tasks.

In summary, the primary contributions of this work are:

- We propose MoPFormer, an effective pre-training framework that achieves state-of-the-art performance.

- We introduce a representation method that enables unified training across heterogeneous datasets.

- We extract motion primitives and provide a detailed interpretability analysis of them.

## 2 Related Work

Traditional human activity recognition approaches train model parameters on different datasets separately, including statistical feature extraction methods and deep learning approaches. Deep learning methods include CNN-based architectures such as CALANet [33], COA-HAR [50], MA-CNN [37], and SenseHAR [24], RNN-based approaches like DeepConvLSTM [30], and attention-based models such as THAT [26] and PA-HAR [51]. Recent research has proposed general-purpose time series models applicable to various tasks, including classification, such as TimesNet [49], TSLANet [17], and FITS [52]. However, these methods are typically trained and tested on splits from the original datasets and are limited in their ability to achieve cross-dataset generalization.

Self-supervised human activity recognition methods perform representation learning through various approaches. Reconstruction-based methods such as TST [55] and TARNet [13] focus on rebuilding input signals. Contrastive learning methods such as TS2Vec [53], CL-HAR [35], DDLearn [36], TS-TCC [16], FOCAL [28], and ModCL [22] execute discriminative representation learning tasks. Other self-supervised objectives like BioBankSSL [55, 14] and Step2Heart [45] also contribute to learning time series representations. These methods typically learn feature representations on specific datasets and fine-tune classifier heads on the target dataset, but they can only perform on datasets with similar structures and still require labeled data from the target dataset.

Research on sensor-based HAR for composite activity recognition is relatively limited [11]. For instance, Chen et al. decomposed composite activities into multiple simple activities [11], where each simple activity is represented by a sequence of sensor signal segments. These segments are first fed into a CNN to extract representations for identifying simple activities. Concurrently, the CNN-extracted features from all segments are passed to an LSTM network to achieve high-level semantic activity classification. Similarly, prior work [12] inferred composite activities using estimated activity sequences, where temporal correlations of simple activities were extracted for composite activity classification. Conversely, predicted composite activities were used to aid the derivation of simple activity sequences for the next time step, with predictions for both simple and composite activity sequences mutually updating during inference. While these studies present viable approaches for composite activity research, datasets with comprehensive, multi-level hierarchical labels for composite activities are scarce, and existing hierarchical annotations are often incomplete.

The concept of motion primitives has gained traction in robotics research [41, 9]. These primitives are fundamental movement patterns that can be sequenced and combined to generate or understand a diverse repertoire of complex human activities. For instance, Calvo et al. utilized eight such primitive movements, including "static" and partial "squat", to train Hidden Markov Models for activity classification. Similarly, Sanzari et al. developed a framework for automatically discovering human motion primitives from 3D pose data by optimizing "motion flux" and then clustering the results [41]. However, the application of motion primitives to enhance the interpretability of IMU-based HAR has seen limited progress. This limitation stems largely from the inherently non-intuitive nature of IMU sensor data, which makes it challenging to define and interpret primitives directly from these signals.

Much of the research on interpretability in HAR has focused on attention mechanisms. The principle behind deep attention models is to weight input components, with the assumption that components assigned higher weights are more relevant to the recognition task and exert greater influence on the model's decisions [43]. For example, Shen et al. [44] designed a segment-level attention method to determine which time periods contain more information; combined with a gated CNN, this segment-level attention can extract temporal dependencies. Zeng et al. [54] developed attention mechanisms from two perspectives: they first proposed sensor attention on the input to extract salient sensory patterns, and then applied temporal attention to LSTMs to filter out inactive data segments. Ma et al. [29] employed spatial and temporal attention mechanisms, extracting spatial dependencies by fusing patterns with self-attention. However, their interpretability often remains at a superficial level, indicating what the model focuses on, rather than why or how these features contribute to the recognition of complex, hierarchically structured activities.

## 3 Methodology

The architecture of MoPFormer, as illustrated in Fig. 2, comprises four modules: the motion primitive module, the context-aware embedding module, and two task heads. The motion primitive module first

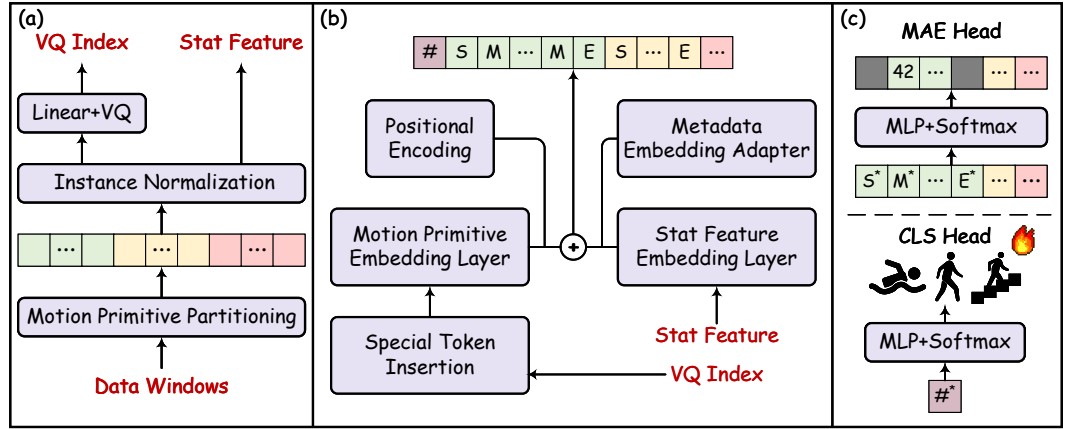

Figure 2: Detailed illustration of key modules in our motion-centric framework. (a) Motion Primitive Module: raw data windows from multiple sensor channels are partitioned into motion primitives and processed through instance normalization to generate Vector Quantization (VQ) indices and statistical features. (b) Context-Aware Embedding Module: special tokens are inserted alongside masked motion embedding tokens ($M$) to form the complete input representation, which combines motion primitive embeddings, positional encodings, and statistical feature embeddings. (c) Task Heads: The diagram shows transformed features $X^*$ representing the corresponding position vectors after Transformer Encoder processing. The MAE head utilizes $M^*$ positions for masked token prediction during pretraining, while the trainable CLS head operates exclusively on the transformed [CLS] token representation for downstream task fine-tuning.

transforms multi-dimensional time-series data windows into index sequences from a fixed vocabulary. To further extract patterns, the context-aware embedding module extracts semantic information from contextual relationships and embeds high-dimensional vector sequences for the Transformer backbone. The Masked Auto-Encoding (MAE) head, as a pre-training task, strengthens the context-aware embedding module's ability to understand motion primitives, while the Classification (CLS) head enables the Transformer backbone to learn effective features for classification. Fig. 2 illustrates the detailed architecture of each key module in our framework.

## 3.1 Motion Primitive Module

The Motion Primitive Module transforms raw data windows into discrete motion primitives, serving as the foundational, interpretable units for our motion-centric representation. For each data window of shape $T \times C$, where $T$ is the number of time steps and $C$ is the number of channels (e.g., accelerometer, gyroscope), we first partition the sequence into non-overlapping segments across each channel independently. Each channel is divided into segments of length $L$, resulting in $S = \lfloor T/L \rfloor$ segments per channel and a total of $S \times C$ segments per window.

Each segment undergoes a feature extraction process where we compute both low-level temporal patterns and statistical characteristics. For each segment $s_i \in \mathbb{R}^L$ (a single-channel segment), we apply instance normalization in Eq. (1) to standardize the input:

$$\hat{s}_i = \frac{s_i - \mu(s_i)}{\sigma(s_i) + \epsilon}, \tag{1}$$

where $\mu(s_i)$ and $\sigma(s_i)$ represent the mean and standard deviation of segment $s_i$, and $\epsilon$ is a small constant for numerical stability. This normalization removes sensor-specific scale differences, ensuring all segments are on a comparable scale before quantization.

The normalized segments are then encoded into discrete motion primitives using a Vector Quantization (VQ) [47] approach. We maintain a learnable codebook $\mathcal{Z} = \{z_1, z_2, ..., z_K\}$, which contains $K$ prototype vectors (or "motion primitives"), each of dimension $L$. As illustrated in Fig. 2(a), for each normalized segment $\hat{s}_i$, we compute its VQ index through Eq. (2) by finding the nearest prototype in the codebook:

$$q_i = \underset{k \in \{1, 2, ..., K\}}{\arg\min} \|\hat{s}_i - z_k\|_2^2. \tag{2}$$

This quantization process maps continuous sensor data to discrete motion primitives, establishing a "vocabulary" of motion primitives. Alongside the VQ indices, we extract statistical features from each segment (mean, variance) to capture complementary information that might be lost during quantization. These statistical features $f_i$ are concatenated with the VQ indices to form the complete motion primitive representation.

The codebook is learned end-to-end through a commitment loss defined in Eq. (3) that encourages consistency between the input segments and their quantized representations:

$$\mathcal{L}_{VQ} = \|sg[\hat{s}_i] - z_{q_i}\|_2^2 + \beta\|\hat{s}_i - sg[z_{q_i}]\|_2^2, \tag{3}$$

where $sg[\cdot]$ denotes the stop-gradient operator and $\beta$ is a hyperparameter controlling the commitment cost.

## 3.2 Context-Aware Embedding Module

The Context-Aware Embedding Module transforms the discrete motion primitive indices and their associated statistical features into rich, contextualized representations suitable for the Transformer backbone. This module aims to capture not only the type of motion primitive but also its intensity, variability, sensor origin, and temporal position. For each data window, the module processes the sequence of VQ indices $\{q_1, q_2, ..., q_S\}$ and corresponding statistical features $\{f_1, f_2, ..., f_S\}$ from all channels.

For the special tokens [MASK], [START], and [END], we assign reserved indices ($K + 1$, $K + 2$, and $K + 3$, respectively) in our vocabulary, ensuring they are consistently represented across different sensor channels. Using a learnable embedding matrix $E_{VQ} \in \mathbb{R}^{(K+3)\times D}$, where $D$ is the embedding dimension, we embed the VQ indices through Eq. (4):

$$e_i^{VQ} = E_{VQ}[q_i]. \tag{4}$$

The [CLS] token is a separate learnable parameter outside the VQ embedding matrix.

The statistical features obtained during instance normalization (mean, variance) are projected into the same embedding space via a linear projection in Eq. (5):

$$e_i^{stat} = W_{stat}f_i + b_{stat}. \tag{5}$$

To incorporate sensor metadata (e.g., sensor type, mounting position) and enhance the model's ability to generalize across different sensor configurations and distinguish between data from different sources, we introduce the Metadata Embedding Adapter for sensor-specific embeddings. For each segment $i$, its corresponding sensor channel $c$ has associated metadata. This metadata is first converted into a fixed-length vector $e_c^{meta} \in \mathbb{R}^N$ using a pre-trained text embedding model. The Metadata Embedding Adapter uses a linear layer ($W_{adapter} \in \mathbb{R}^{D_{model}\times N}$, $b_{adapter} \in \mathbb{R}^{D_{model}}$), then maps this $N$-dimensional vector to our model's embedding dimension $D_{model}$.

For each segment $i$, its complete token embedding $e_i$ is formed by summing these constituent embeddings, as shown in Eq. (6). This fusion represents the motion primitive in a way that includes its quantized shape ($e_i^{VQ}$), magnitude/variability information ($e_i^{stat}$), and sensor context ($e_c^{adapter\_emb}$):

$$e_i = e_i^{VQ} + e_i^{stat} + W_{adapter}(e_c^{meta}) + b_{adapter}, \tag{6}$$

where $e_c^{meta}$ is the pretrained text embedding of dimension $N$ for the channel $c$ that segment $i$ belongs to.

The model incorporates various special tokens to provide structural cues in the input sequence.

The final input sequence to the transformer backbone is structured as in Eq. (7):

$$X = [[CLS], [START], e_1^1, ..., e_{n_1}^1, [END], ..., [START], e_1^C, ..., e_{n_C}^C, [END]], \tag{7}$$

where $e_j^c$ represents the $j$-th embedding from sensor channel $c$, and $n_c$ is the number of segments for channel $c$.

To provide temporal context, positional encodings are added through Eq. (8):

$$\hat{X} = X + P, \tag{8}$$

where $P \in \mathbb{R}^{S\times D}$ contains learnable position embeddings. Importantly, our positional encoding scheme assigns identical positional embeddings to data from different channels at the same time step, meaning that $e_1^1$ and $e_1^C$ receive the same positional encoding if they represent data from the same time point but different channels.

### 3.3 Task Heads

As shown in Fig. 2(c), MoPFormer employs a dual-task learning approach with two specialized heads built on top of the transformer encoder:

**Masked Auto-Encoding (MAE) Head.** During pre-training, we randomly mask a portion of the motion embeddings in the input sequence and replace them with [MASK] tokens. The MAE head aims to reconstruct the original VQ indices of these masked positions based on the contextual information processed by the transformer encoder. Eq. (9) shows how for each masked position $i$, the reconstruction is performed:

$$\hat{q}_i = \text{softmax}(W_{mae}h_i^* + b_{mae}), \tag{9}$$

where $h_i^*$ is the transformer output at position $i$, and $W_{mae}$ and $b_{mae}$ are learnable parameters. The MAE objective in Eq. (10) is formulated as a cross-entropy loss between the predicted and true VQ indices:

$$\mathcal{L}_{mae} = -\frac{1}{|M|}\sum_{i \in M} \log \hat{q}_i[q_i], \tag{10}$$

where $M$ is the set of masked positions and $\hat{q}_i[q_i]$ is the probability assigned to the true index $q_i$.

**Classification (CLS) Head.** For downstream activity recognition tasks, we utilize a classification head that operates exclusively on the transformed [CLS] token representation. The [CLS] token aggregates information from the entire sequence through self-attention mechanisms in the transformer encoder. Eq. (11) defines how the classification is performed:

$$\hat{y} = \text{softmax}(W_{cls}h_{cls}^* + b_{cls}), \tag{11}$$

where $h_{cls}^*$ is the transformed [CLS] token representation and $W_{cls}$ and $b_{cls}$ are learnable parameters. The classification objective is formulated as a cross-entropy loss in Eq. (12):

$$\mathcal{L}_{cls} = -\sum_{j=1}^{C} y_j \log \hat{y}_j, \tag{12}$$

where $y$ is the true activity label. During pre-training, we primarily focus on the MAE task, while during different training phases, we balance various objectives through Eq. (13):

$$\mathcal{L} = \lambda_{mae}\mathcal{L}_{mae} + \lambda_{cls}\mathcal{L}_{cls} + \lambda_{vq}\mathcal{L}_{vq}, \tag{13}$$

where $\lambda_{mae}$, $\lambda_{cls}$, and $\lambda_{vq}$ are hyperparameters balancing the different loss components from Eqs. (10), (12), and (3), respectively. By adjusting these hyperparameters, we can control which tasks are optimized during different training stages.

For fine-tuning on downstream tasks, we freeze the Motion Primitive Module and Context-Aware Embedding Module while only updating the classification head parameters, enabling efficient adaptation to new datasets with minimal computational overhead.

## 4 Experiments

### 4.1 Experimental Setup

**Datasets.** We conducted extensive evaluations using six publicly available benchmark datasets: PAMAP2 [38, 39], DSADS [7, 1], MHealth [5, 6], Realworld [46], UCI-HAR [40, 2], and USC-HAD [56]. To ensure fair comparison with existing self-supervised methods, we pre-trained our model on five of these datasets while using the remaining dataset for evaluation, maintaining consistent experimental settings with other self-supervised approaches. All datasets are resampled to 100 Hz and segmented with a 500-sample window. To eliminate information leakage from overlapping windows, we adopt the setting by using strides equal to the window size for all labeled training segments. This strict non-overlapping approach may reduce absolute performance compared with more permissive settings, but ensures a more rigorous evaluation. We use accuracy and macro-F1 as evaluation metrics to assess model performance.

Table 1: Comparison with supervised pretrain-and-transfer baselines (upper block) and supervised train-from-scratch baselines (middle block) across six HAR datasets. Accuracy and macro-F1 are reported in percent; the rightmost column lists the mean over all datasets.

| Method | | PAMAP2 | DSADS | MHealth | Realworld | UCI-HAR | USC-HAD | Average |
|---|---|---|---|---|---|---|---|---|
| BYOL [21]+perm_jit | Acc | 80.88 | 91.30 | 88.94 | 85.18 | 82.66 | 74.04 | 83.83 |
| | F1 | 79.01 | 89.22 | 88.59 | 86.81 | 80.32 | 73.92 | 82.98 |
| BYOL [21]+lfc | Acc | 76.76 | 90.80 | 89.74 | 86.08 | 81.18 | 72.82 | 82.90 |
| | F1 | 74.64 | 89.39 | 88.84 | 85.99 | 80.87 | 69.23 | 81.49 |
| BYOL [21]+Mixup | Acc | 82.38 | 93.87 | 90.06 | 88.80 | 85.31 | 75.59 | 86.00 |
| | F1 | 80.93 | 93.29 | 89.32 | 87.95 | 86.78 | 70.04 | 84.72 |
| ModCL [22] | Acc | 82.49 | 92.46 | 90.19 | 89.69 | 89.79 | 76.64 | 86.88 |
| | F1 | 80.63 | 91.09 | _89.98_ | 90.43 | **90.15** | 73.61 | 85.98 |
| TSLANet [17] | Acc | **86.76** | **98.12** | 89.84 | 90.03 | **91.02** | _79.93_ | _89.28_ |
| | F1 | _84.08_ | **97.43** | 87.85 | _91.81_ | _89.84_ | **77.74** | _88.13_ |
| CALANet [34] | Acc | 85.10 | 96.01 | 86.05 | 90.07 | 88.38 | 77.37 | 87.16 |
| | F1 | 83.94 | 95.58 | 83.80 | 91.59 | 86.91 | 72.36 | 85.70 |
| MoPFormer | Acc | _86.08_ | _97.60_ | **93.22** | **92.05** | _90.58_ | **81.46** | **90.17** |
| | F1 | **84.83** | _97.38_ | **92.28** | **93.25** | _89.98_ | _77.67_ | **89.23** |
| w/o Pretrain | Acc | 85.76 | 94.68 | _90.53_ | _91.84_ | 77.58 | 78.30 | 86.45 |
| | F1 | 83.94 | 94.64 | 89.95 | 91.33 | 76.79 | 74.86 | 85.25 |

**Training Protocol.** All experiments were conducted on a Quadro RTX 8000 GPU. Our training protocol followed a two-stage approach. In the first pre-training stage, we trained the model across multiple datasets to extract a diverse set of motion primitives. We set the segment size to 50 samples for Motion Primitive length, used an internal embedding dimension of 256, and masked 25% of motion primitives as prediction targets for the MAE task. In the second stage, the classification head and transformer layers are fine-tuned on each downstream dataset.

**Metadata Embedding Adapter.** Sensor descriptors are standardized to a consistent string format (e.g., "body_part: Chest, sensor: acc, axis: x"). We then obtained embeddings with Google's text-embedding-004 API, which enables the model to learn from contextual sensor metadata effectively.

## 4.2 Results

**Baseline Comparisons.** We compared MoPFormer with six different baseline approaches. Following the comprehensive evaluation in [35], we selected the BYOL framework [21], which demonstrated the best average performance among various contrastive learning frameworks. For time-domain and frequency-domain augmentations, we used perm_jit and lfc, respectively, which showed superior performance in their domains. As noted in [15], Mixup provides distinctive benefits compared to other temporal augmentation methods, so we included it as a baseline. ModCL [22] represents a state-of-the-art contrastive learning method that leverages both intra-modal and inter-modal consistency in wearable sensor data. We included it as a contemporary contrastive learning approach. For all contrastive learning methods, we maintained a consistent 0.2:0.8 train-test split ratio.

Additionally, we compared against recent supervised approaches, including TSLANet [17] (ICML 2024), a widely recognized foundational model in the time-series domain, and CALANet [34] (NeurIPS 2024), which was specifically designed for HAR tasks [8]. Detailed parameter configurations for all baseline models are provided in Appendix A.

**Performance Analysis.** Tab. 1 presents comprehensive results across all six datasets. MoPFormer achieves state-of-the-art performance, obtaining the highest average accuracy and F1 scores. Specifically, our model achieved the best performance on MHealth (93.22% Accuracy, 92.28% F1) and USC-HAD (81.46% Accuracy), while maintaining competitive results on other datasets.

The ablation study (w/o pretrain) demonstrates that our pretraining strategy significantly contributes to model performance, particularly on datasets like UCI-HAR with limited data but complex activity patterns, where performance drops by 13 percentage points without pretraining. This confirms the value of our two-stage training approach with motion primitive extraction and masked autoencoding.

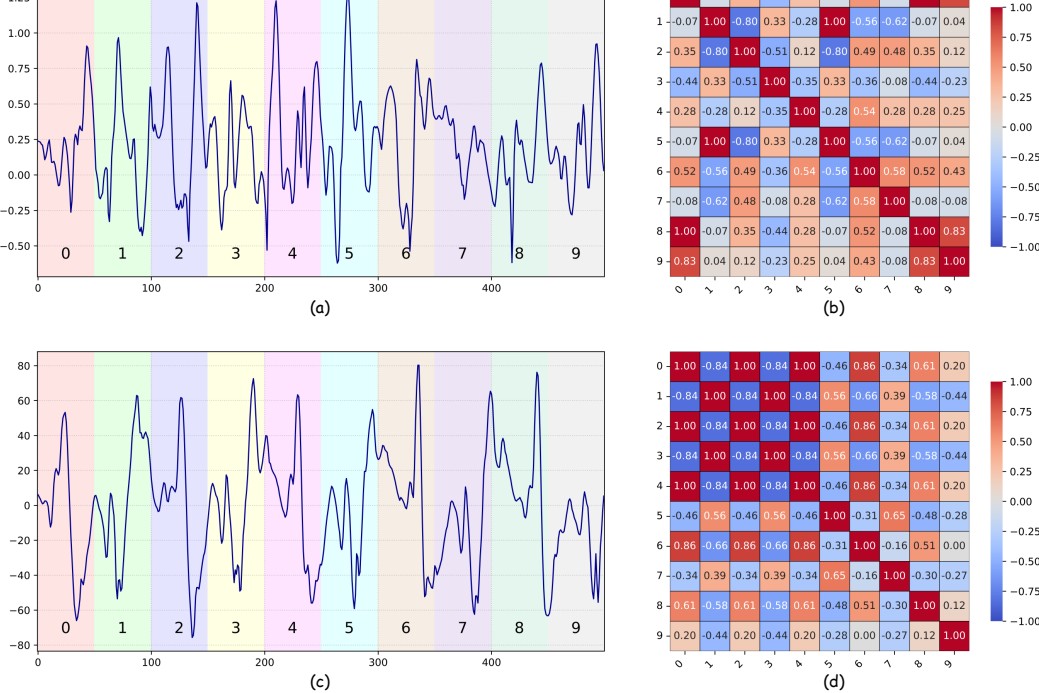

Figure 3: Motion primitive segmentation and similarity analysis. (a) 5-second raw accelerometer trace from USC-HAD dataset, segmented into ten 0.5-second motion primitives. (b) Cosine-similarity matrix of motion primitive embeddings from accelerometer data. (c) Corresponding 5-second gyroscope trace with identical segmentation. (d) Cosine-similarity matrix of embeddings for gyroscope-based motion primitives. The matrices reveal pattern correlations between different motion primitives after Motion Primitive Embedding processing.

**Ablation Study.** To validate each component's contribution, we conduct an ablation study on two representative datasets (Tab. 2). First, removing the pretraining stage leads to a modest accuracy drop, highlighting the importance of motion-primitive initialization in cross-modal generalization. Next, further omitting the statistical features or excluding metadata embedding causes a dramatic performance loss.

Table 2: Ablation study for each component on the PAMAP2 and DSADS datasets.

| Method | PAMAP2 | | DSADS | |
|---|---|---|---|---|
| | Acc | F1 | Acc | F1 |
| MoPFormer | 86.08 | 84.83 | 97.60 | 97.38 |
| - w/o Pretrain | 85.76 | 83.94 | 94.68 | 94.64 |
|   - w/o Statistical Feature | 69.95 | 53.60 | 71.15 | 62.34 |
|   - w/o Metadata Embedding | 58.18 | 53.10 | 80.94 | 75.40 |

Together, these findings show that pretraining, statistical features, and metadata embeddings each deliver unique information, and that the combination is critical to MoPFormer's superior performance. More results are put in Appendix B.

MoPFormer's strong performance across diverse datasets validates our motion-centric approach and demonstrates its generalization capabilities. In the following section, we will further analyze the extracted motion primitives better to understand their contribution to the model's effectiveness.

## 5 Analysis

### 5.1 Motion Primitive Similarity

To examine what the VQ codebook captures, we analyzed two example segments of five seconds each, shown in Fig. 3. Each trace is divided into ten 0.5s motion primitives, and we compute pairwise cosine similarity between the embeddings of those motion primitives. A consistent pattern emerges: primitives describing similar kinematic shapes, such as indices 8 and 9 in (a) or indices 1 and 3 in

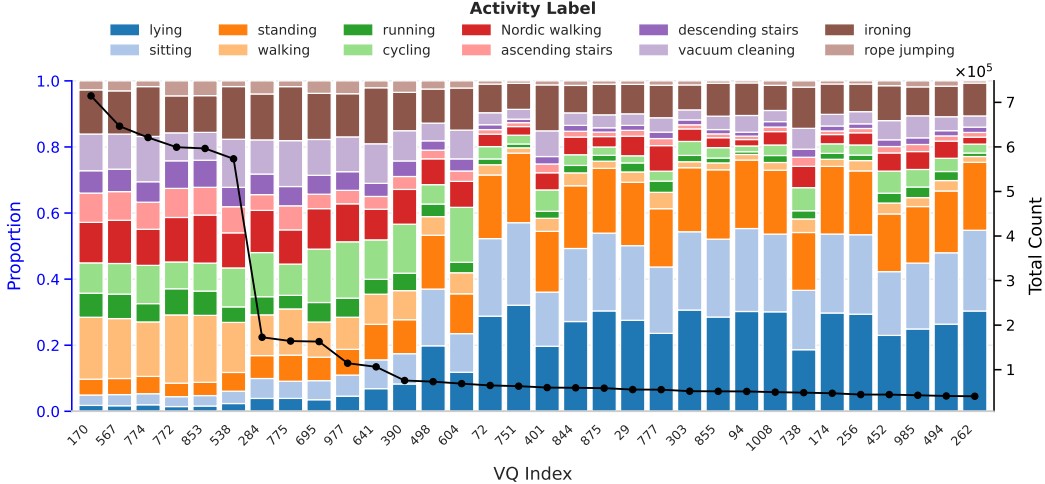

Figure 4: Frequency and activity composition of the 32 most common motion primitives from PAMAP2. The stacked bars (left axis) show the proportion of each activity label for every VQ index, while the black line (right axis) plots the absolute occurrence count of motion primitives.

(c), cluster into blocks with high cosine similarity. In contrast, primitives with opposite or dissimilar trends (e.g., indices 1 and 3 in (a), and indices 0 and 1 in (c)) exhibit low similarity.

## 5.2 Motion Primitive Frequency

To further explain the rationale of the learned codebook, we analyzed both the overall frequency of each primitive and its distribution across activity classes (using the PAMAP2 dataset for illustration). Fig. 4 presents the 32 most frequent VQ indices: each stacked bar depicts the relative share of 12 activity classes, and the black curve shows the absolute number of occurrences. Because any macroscopic activity can be decomposed into a sequence of fine-grained motion primitives, the codebook naturally captures micro-movements that are reused across different activities. The resulting distribution is highly skewed: the ten most common primitives cover the vast majority of windows, whereas the remaining indices form a long-tailed set that represents rare or transitional motions. Notably, locomotion classes (e.g., walking, running, and cycling) dominate the high-frequency primitives (e.g., primitive #170 and primitive #567), while low-variance postures such as lying, sitting, and standing are scattered among the less frequent primitives (e.g., primitive #72 and primitive #751). This frequency analysis could allow us to identify which primitive motions are more common and how they relate to high-level activities, which is a step toward explaining model decisions.

## 6 Conclusion and Future Work

In this work, we introduced MoPFormer, a motion-primitives-based Transformer framework for wearable-sensor activity recognition. Our approach tackles two key HAR challenges: interpretability and cross-domain generalization. Through comprehensive experiments, we showed that MoPFormer achieves state-of-the-art classification performance on six diverse datasets, outperforming both supervised and self-supervised baselines. We also demonstrated that MoPFormer's learned primitives are semantically meaningful. This interpretability lets us peek inside the "black box" of the HAR model. Overall, MoPFormer combines the strengths of sequence modeling and symbolic representation learning, yielding a HAR system that is both accurate and explainable.

Our future work will focus on two key directions. First, developing more robust foundational models by pre-training on larger, more diverse sensor datasets to extract richer motion primitives. Second, aligning our learned motion primitive embeddings with large language models as a novel input modality enables more flexible activity recognition and potentially expresses transfer ability.

## Acknowledgments

This work was supported by National Key Research and Development Program of China (No. 2022YFA1004102) and NSFC grant (No. 62136005 and 62202455).

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

# A    Experimental Settings

In this section, we provide detailed parameter settings for our proposed MoPFormer model and baseline models used in our experiments. All experiments can run normally on the RTX 8000 with 40GB VRAM (requiring the use of gradient accumulation techniques).

## A.1    Parameter Settings for MoPFormer

We implement the MoPFormer model with the following configuration. For motion representation, we use 50 as the motion primitive length, which corresponds to 0.5s of motion when resampled to 100Hz. The codebook size is set to 1024 with an embedding dimension of 256. The architecture consists of 5 standard Transformer Encoder layers with 8 attention heads per layer and an MLP ratio of 1. We use GELU as the activation function throughout the network. Notably, we do not employ any dropout in our model. During training, we apply a masking ratio of 0.25 for the masked modeling objective. For optimization, we employ the AdamW optimizer with a learning rate of 1e-4 and a weight decay coefficient of 1e-5. We use a batch size of 512, implemented with gradient accumulation using PyTorch Lightning to optimize memory usage.

## A.2    Parameter Settings for Baseline Models

**BYOL**   For BYOL [21] (Bootstrap Your Own Latent), we implement the model following the CL-HAR [35] library's configuration. The architecture consists of two networks: an online network and a target network, both utilizing the DCL (DeepConvLSTM) architecture as the backbone encoder. The online network includes a projection head that maps the backbone's output to a 128-dimensional embedding space with a hidden layer of the same dimension, structured as a sequence of Linear, BatchNorm1d, ReLU, Linear, and BatchNorm1d layers. The prediction head follows with the same dimensionality (128) and is structured as Linear, BatchNorm1d, ReLU, and Linear layers. The target network's parameters are updated via an exponential moving average of the online network's parameters with a decay rate of 0.996. We employ three different data augmentation techniques, all following the settings provided in [35]. The model is trained using negative cosine similarity as the loss function with the Adam optimizer. We set the learning rate to 1e-4 for the online encoder and 1e-3 for the online predictor, with a weight decay of 1.5e-6, using a CosineAnnealingLR scheduler. For downstream activity recognition, we freeze the backbone encoder and train a linear classifier with the Adam optimizer at a learning rate of 1e-3 for the same number of epochs.

**ModCL**   For ModCL, we follow the default parameter settings as described in the original paper. Since the official code was not publicly available, we implemented the model based on the paper's specifications. During our reproduction, we enabled several data augmentation techniques, specifically applying five methods: Jittering, Scaling, Permutation, Masking, and Time Warping, all using the same proportions and parameters as specified in the original work.

**TSLANet**   For TSLANet, we implement the model using the default parameter settings specified in the original paper. For parameters not explicitly mentioned in the paper, we adopt the default configurations from the authors' publicly released code repository. Following the original architecture, we enable both the ICB (Input Channel Block) and ASB (Attention Sub-Block) modules. The model is trained with the same optimization strategy and hyperparameters as described in the original TSLANet paper.

**CALANet**   For CALANet, we follow the default parameter settings described in the original paper. For configuration details not explicitly stated in the paper, we use the default settings provided in the authors' official code implementation. We set the number of layers in the layer aggregation pool to 9 based on the authors' ablation study results, which identified this as the optimal configuration. All other hyperparameters and training procedures follow the specifications in the original paper.

# B    Supplementary Ablation Studies

Since our work primarily focuses on investigating interpretable motion primitives in human activity recognition (HAR), the main paper demonstrates the necessity of each architectural component

through module-wise ablation experiments. This appendix provides additional ablation studies examining the impact of key hyperparameters on model performance to provide comprehensive insights into our method's design choices.

## B.1 Motion Primitive Window Length

The choice of motion primitive window length significantly affects the model's ability to capture meaningful motion patterns. We evaluate different window lengths to determine the optimal temporal granularity for motion primitive extraction.

Table 3: Ablation study on motion primitive window length

| Window Length | PAMAP2 | | DSADS | |
|---|---|---|---|---|
| | Acc | F1 | Acc | F1 |
| 25 | 85.71 | 82.11 | 97.09 | 96.37 |
| 50 (ours) | 86.08 | 84.83 | 97.60 | 97.38 |
| 100 | 80.81 | 79.85 | 92.94 | 92.17 |

As shown in Table 3, a window length of 50 achieves optimal performance across both datasets. While shorter windows 25 demonstrate comparable performance, we select the length of 50 to ensure sufficient temporal context for comprehensive motion primitive analysis. This choice prioritizes capturing complete motion phases within each primitive, which is crucial for meaningful interpretability analysis. Longer windows 100 lead to substantial performance degradation due to the inclusion of multiple distinct motion phases within a single primitive, confirming that our selected window length provides the appropriate temporal granularity without compromising motion primitive coherence.

## B.2 Transformer Encoder Depth

We investigate the impact of transformer encoder depth on the model's representational capacity and its effect on motion primitive quality.

Table 4: Ablation study on transformer encoder layer number

| Depth | PAMAP2 | | DSADS | |
|---|---|---|---|---|
| | Acc | F1 | Acc | F1 |
| 2 | 85.01 | 84.59 | 96.92 | 96.56 |
| 5 (ours) | 86.08 | 84.83 | 97.60 | 97.38 |
| 7 | 86.11 | 84.98 | 97.53 | 97.31 |
| 10 | 86.05 | 84.79 | 97.28 | 97.02 |

The results in Table 4 show that performance differences between 5, 7, and 10 layers are minimal. However, we adopt 5 layers as our configuration to ensure adequate modeling capacity for motion primitive extraction without introducing unnecessary complexity. This choice reflects our preference for slightly conservative parameter settings that guarantee sufficient representational power for motion primitive analysis, while avoiding potential overfitting. The 2-layer configuration shows reduced performance, confirming that our selected depth provides the necessary capacity for high-quality motion primitive learning.

## B.3 Hidden Embedding Dimensionality

The dimensionality of hidden embeddings critically determines the model's representational capacity for motion primitive encoding and subsequent interpretability analysis.

From Table 5, we observe that performance reaches saturation around 128 dimensions, with 256 dimensions providing marginal improvements. We choose 256 dimensions to ensure comprehensive representational capacity that does not limit our motion primitive analysis. This decision follows our principle of using slightly excessive parameters to guarantee that motion primitive quality and

Table 5: Ablation study on hidden embedding size

| Embedding Size | PAMAP2 | | DSADS | |
|---|---|---|---|---|
| | Acc | F1 | Acc | F1 |
| 32 | 82.18 | 81.70 | 94.81 | 93.88 |
| 64 | 85.90 | 85.86 | 96.93 | 96.31 |
| 128 | 86.45 | 85.50 | 97.28 | 97.35 |
| 256 (ours) | 86.08 | 84.83 | 97.60 | 97.38 |

interpretability are not compromised by insufficient representational capacity. Lower dimensions (32-64) show significant performance degradation, validating that our selected dimensionality provides adequate space for capturing the complexity and nuances of human motion patterns.

## B.4 Vector Quantization Codebook Size

The VQ codebook size determines the discrete vocabulary available for motion primitive representation, directly impacting both model performance and the richness of discoverable motion primitives.

Table 6: Ablation study on VQ codebook size

| Codebook Size | PAMAP2 | | DSADS | |
|---|---|---|---|---|
| | Acc | F1 | Acc | F1 |
| 16 | 78.81 | 76.89 | 90.05 | 88.93 |
| 64 | 84.31 | 83.79 | 95.55 | 95.43 |
| 256 | 85.62 | 84.87 | 97.52 | 97.18 |
| 1024 (ours) | 86.08 | 84.83 | 97.60 | 97.38 |

As demonstrated in Table 6, performance steadily improves with larger codebook sizes and appears to saturate around 256 codes for the current pretraining datasets. However, we adopt 1024 codes to ensure comprehensive coverage of motion primitive diversity, particularly to avoid any limitations that might affect our detailed motion primitive analysis. This choice exemplifies our approach of using generously-sized parameters to guarantee that the learned motion primitives fully capture the richness and subtlety of human motion patterns without being constrained by codebook capacity. Smaller codebooks (16-64) exhibit substantial performance degradation, confirming the necessity of adequate representational diversity for meaningful motion primitive discovery.

## B.5 Model Stability across Random Seeds

To validate the stability of MoPFormer's performance, we conducted additional experiments by fine-tuning our pre-trained model three times with different random seeds on the PAMAP2, DSADS, and RealWorld datasets. For each run, we followed the exact same fine-tuning protocol as in the main experiments (i.e., freezing the pre-trained encoder and training a linear classifier).Table 7 reports the mean and standard deviation of accuracy and F1-score across the three independent runs.

Table 7: Comparison of original reported results with new results from multiple runs (Mean±Std).

| Dataset | Original Reported | | New Results (Mean±Std) | |
|---|---|---|---|---|
| | Acc (%) | F1 (%) | Acc (%) | F1 (%) |
| PAMAP2 | 86.08 | 84.83 | 87.49±4.10 | 85.74±3.04 |
| DSADS | 97.60 | 97.38 | 97.27±0.49 | 97.11±0.24 |
| RealWorld | 92.05 | 93.25 | 91.32±1.07 | 92.65±1.23 |

# C Additional Motion Primitive Analysis

To provide deeper insights into the interpretability of our learned motion primitives, we analyze the distribution of motion primitive usage across different human activities. This analysis demonstrates how distinct activities exhibit characteristic motion primitive patterns, validating the semantic meaningfulness of our learned representations.Figures 5 through 12 present the motion primitive usage distributions for eight representative activities from our dataset.

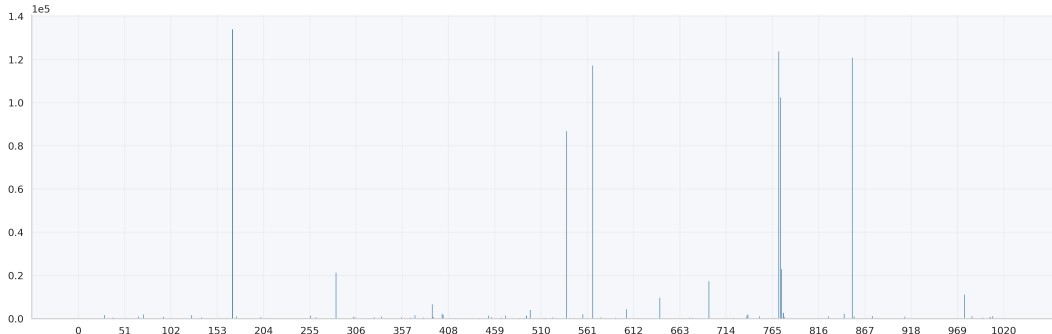

Figure 5: Motion primitive usage distribution for ascending stairs activity.

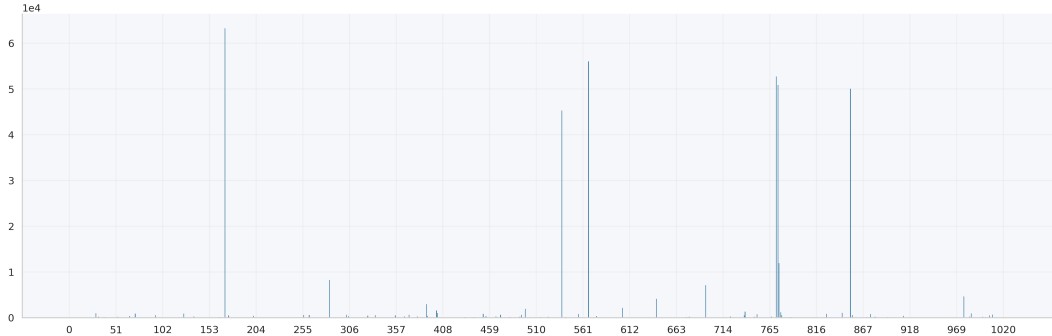

Figure 6: Motion primitive usage distribution for cycling activity.

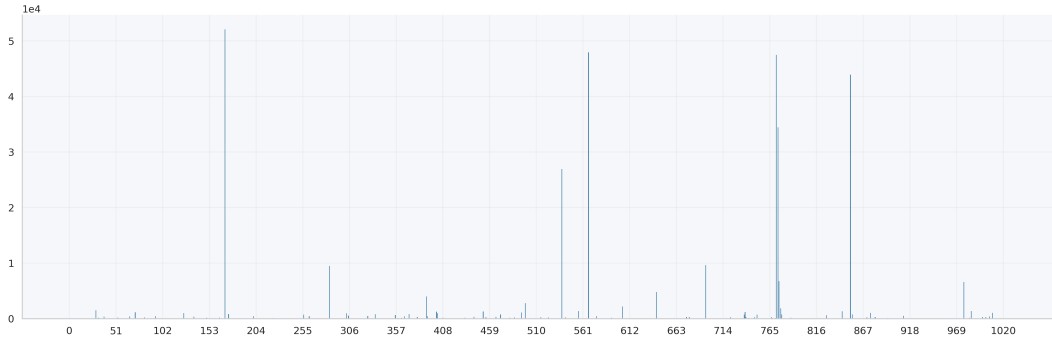

Figure 7: Motion primitive usage distribution for walking activity.

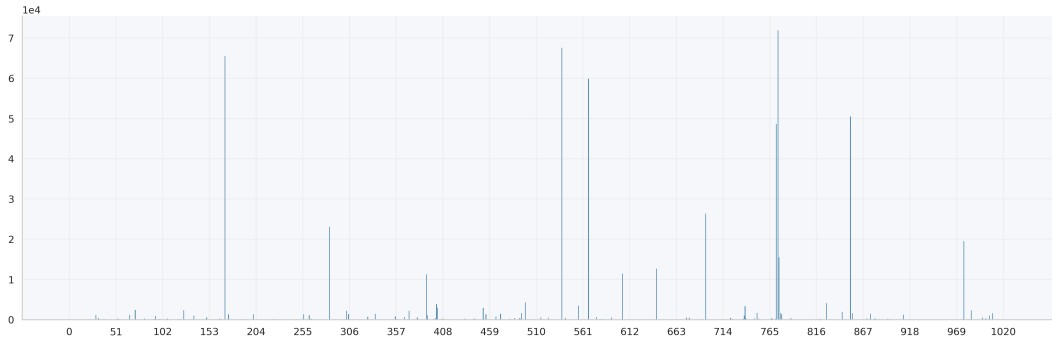

Figure 8: Motion primitive usage distribution for running activity.

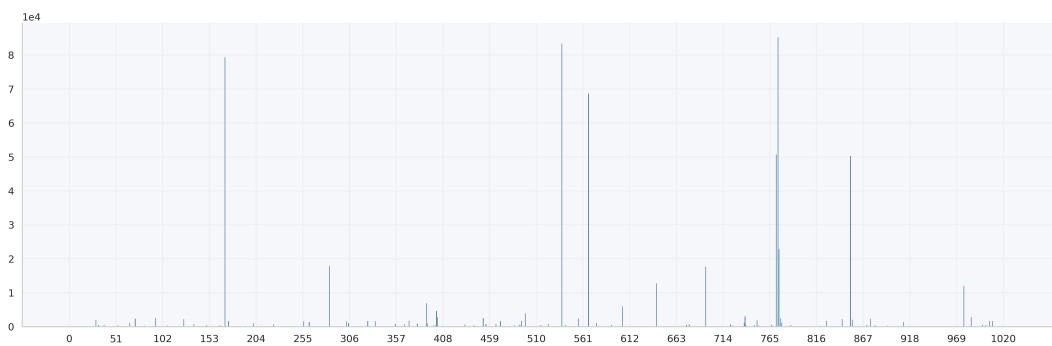

Figure 9: Motion primitive usage distribution for jumping activity.

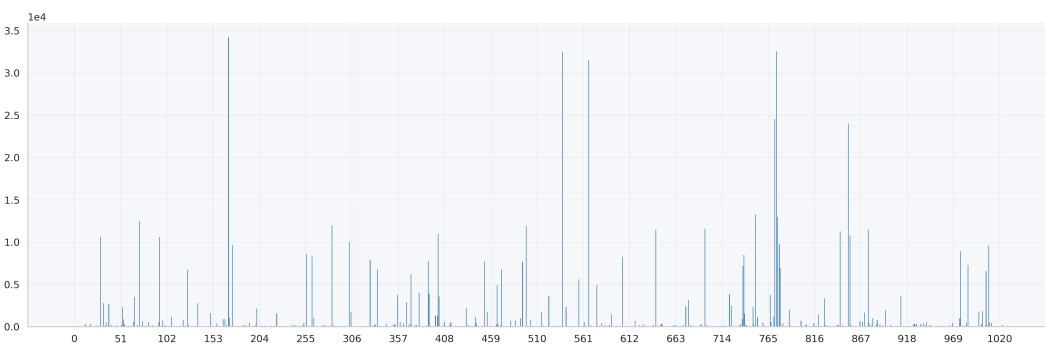

Figure 10: Motion primitive usage distribution for lying activity.

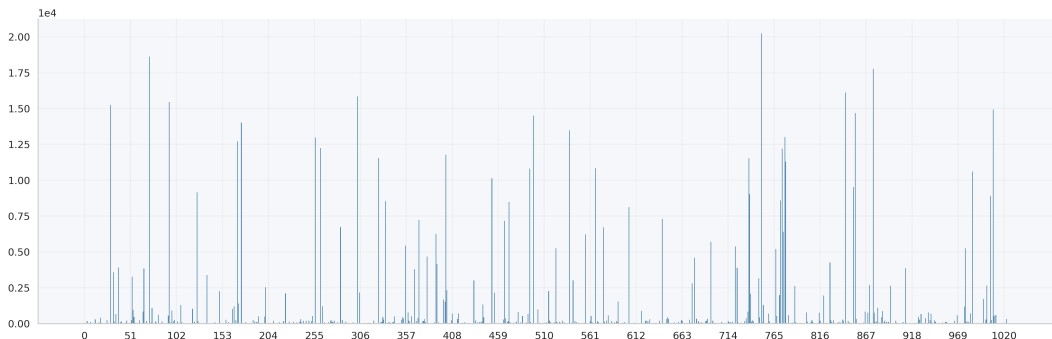

Figure 11: Motion primitive usage distribution for sitting activity.

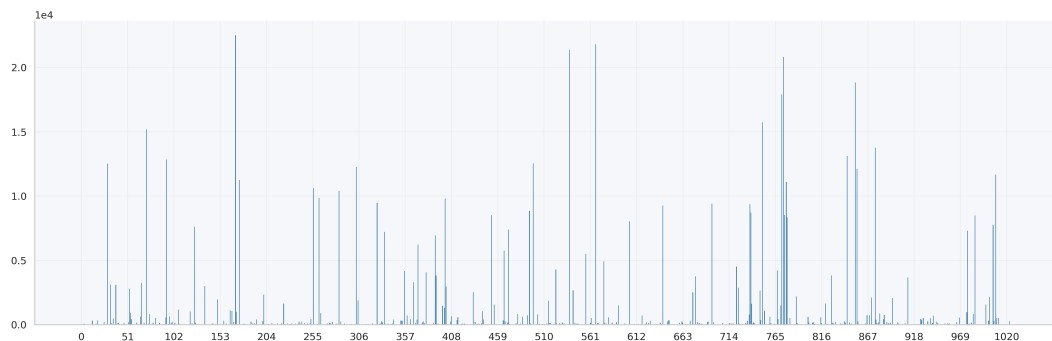

Figure 12: Motion primitive usage distribution for standing activity.

Through careful observation of these distributions, we can identify distinct patterns that reflect the fundamental differences in motion primitive composition across various activities. The distributions clearly separate into two categories: dynamic activities (Figures 5 through 9) and static activities (Figures 10 through 12). This contrast is particularly striking in terms of primitive usage patterns. Dynamic activities demonstrate concentrated utilization of specific motion primitives, with certain primitives exhibiting significantly higher activation frequencies, indicating the presence of characteristic movement patterns essential for these activities. In contrast, static activities show a more uniform distribution across motion primitives, suggesting these behaviors rely on a broader range of subtle motion components for postural maintenance and minor adjustments. This fundamental difference in primitive usage validates that our learned motion primitives capture meaningful biomechanical distinctions between different categories of human activities.

Furthermore, to analyze the temporal relationships between motion primitives, we construct and visualize the Markov transition probability matrix for all learned motion primitives, as illustrated in Figure 13.

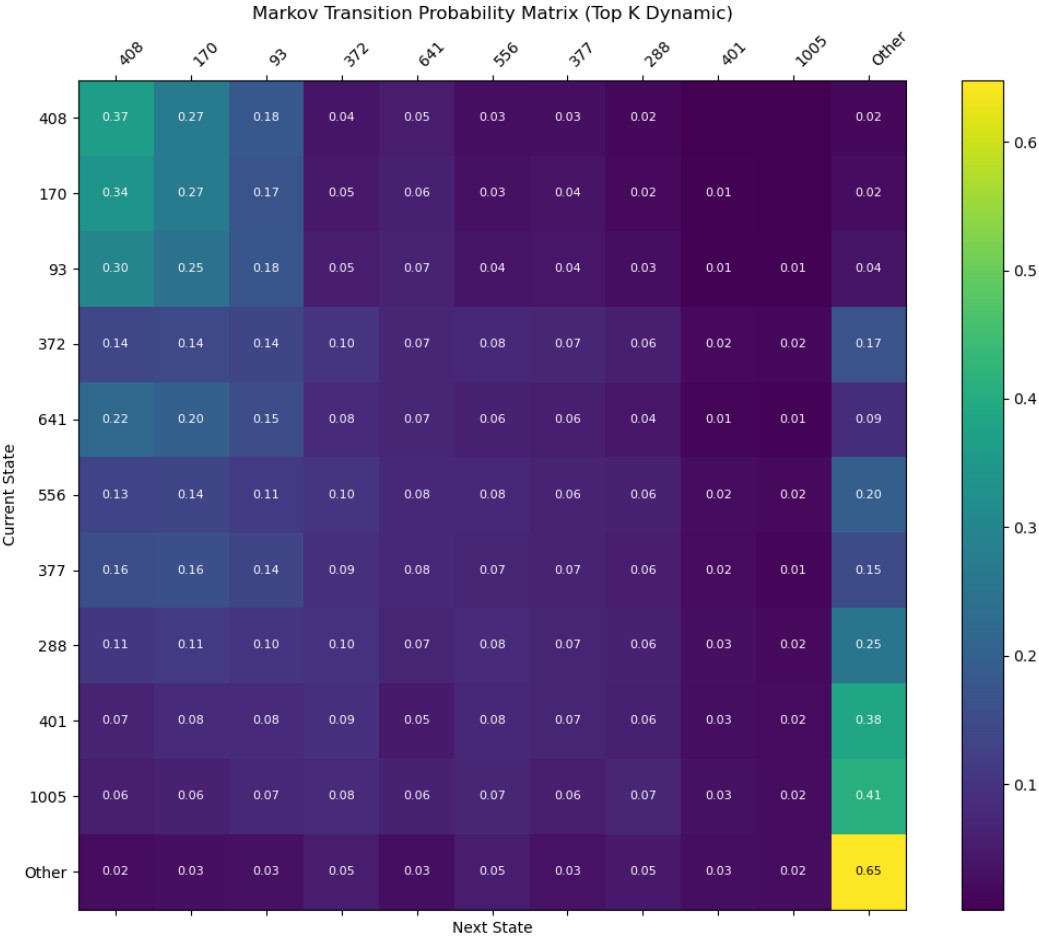

Figure 13: Markov transition probability matrix showing the temporal dependencies between motion primitives.

## D   Dataset Description

All datasets used in this paper adopt a sliding window approach with a window size of 500 samples and a step size of 500 samples as the basic recognition unit. All sensor data is resampled to 100 Hz to ensure consistency across different datasets. Data segments shorter than the specified window size are discarded from the analysis. For fine-tuning and evaluation purposes, we allocate 20% of each

dataset for fine-tuning while the remaining data is used for testing. The datasets employed in this study are described as follows:

**PAMAP2**   The PAMAP2 Physical Activity Monitoring dataset is a comprehensive collection designed for activity recognition and intensity estimation research, containing recordings of 18 different physical activities such as walking, cycling, and playing soccer performed by 9 subjects. Each subject wore three Colibri wireless inertial measurement units (IMUs) positioned on the wrist of the dominant arm, chest, and ankle of the dominant side, sampling at 100 Hz, along with a heart rate monitor operating at approximately 9 Hz. The data collection protocol required each subject to perform 12 standardized activities, with some subjects also completing additional optional activities.

**DSADS**   This dataset captures activity recognition data from 19 different physical activities performed by 8 subjects (4 female, 4 male, aged 20-30). Activities include basic postures, locomotion, daily activities, and exercises, each performed for 5 minutes and segmented into 5-second intervals. Five sensor units were placed on different body locations (torso, arms, legs), with each unit containing accelerometer, gyroscope, and magnetometer sensors. Data was recorded at 25 Hz and organized hierarchically by activity, subject, and segment.

**MHealth**   This dataset contains human activity recognition data from 12 physical activities (stationary postures, locomotion, and exercises) performed by 10 volunteers in natural settings. Three wearable sensors placed on the chest, right wrist, and left ankle measure motion parameters (acceleration, rotation, magnetic orientation), with the chest sensor also capturing ECG data. All recordings were sampled at 50 Hz, providing comprehensive movement and physiological data for activity recognition research.

**Realworld**   This dataset captures human activity recognition data from 8 different physical activities (walking, running, sitting, standing, lying, stairs up, stairs down, jumping) performed by 15 subjects (8 males, 7 females; age 31.9±12.4, height 173.1±6.9 cm, weight 74.1±13.8 kg). Each activity was performed for approximately 10 minutes per subject (except jumping at ∼1.7 minutes due to physical exertion), with data equally distributed between genders. The dataset includes IMU sensor readings (acceleration, gyroscope, magnetic field) collected at a 50 Hz sampling rate simultaneously from 7 different body positions (chest, forearm, head, shin, thigh, upper arm, and waist).

**UCI-HAR**   This dataset captures human activity recognition data from 6 different physical activities (walking, walking upstairs, walking downstairs, sitting, standing, lying) performed by 30 volunteers aged 19-48 years. The experiments were video-recorded for manual data labeling. The sensor signals were pre-processed with noise filters, while a Butterworth low-pass filter with 0.3 Hz cutoff frequency was applied to separate body acceleration from gravity, with features extracted from both time and frequency domains for activity recognition analysis.

**USC-HAD**   This dataset captures human motion data using MotionNode sensors operating at 100Hz (±6g accelerometer range, ±500dps gyroscope range). It includes 12 different physical activities: Walking Forward/Left/Right, Walking Upstairs/Downstairs, Running Forward, Jumping Up, Sitting, Standing, Sleeping, and Elevator Up/Down.

## E   Limitations

Our approach introduces computational overhead through Vector Quantization and multi-layer Transformer processing, which could pose some problem for real-time inference on resource-limited wearable devices.

