# OpenReview forum: "MoPFormer: Motion-Primitive Transformer for Wearable-Sensor Activity Recognition"
_NeurIPS.cc/2025/Conference — NeurIPS 2025 poster_

### Official Review · Reviewer_nWUi · 2025-06-22

**Clarity:** 2
**Significance:** 3
**Originality:** 2
**Rating:** 4
**Confidence:** 4

**Summary:**

In this paper, the authors aim to address 2 challenges in sensor-based human activity recognition, including interpretability and cross-dataset generalization capability. To address these challenges, the authors design a pre-training-based method call MoPFormer. The key idea is the pretraining will improve the performance. Evaluations are conducted on 6 HAR datasets, and results show the proposed MoPFormer have achieved better performance compared to other methods.

**Questions:**

I have questions regarding the experiments, the motivations, etc. Please see the comments above.

**Ethical Concerns:**

["NO or VERY MINOR ethics concerns only"]

**Final Justification:**

The authors have addressed my concerns during the rebuttal process. I have increased the score.

**Limitations:**

Yes

**Paper Formatting Concerns:**

The format is good.

**Quality:**

2

**Strengths And Weaknesses:**

Pros:

1.Overall ,the target problem is interesting. The focused two challenges are practical and could have broader impact.

2.Most figures and tables are well-designed.

3.I like the way the authors present the problem, also, I believe generalization is one of the most urgent challenges in the field. While I am not sure about the interpretabiltiy, it seems an interesting  point to consider with more justifications and analysis.

Concerns:


a. Categorization and Limited Related Work. The reviewer suggests the authors to present related studies in better categorization. Also, there has been multiple cross-dataset, cross-domain human activity recognition papers, I would suggest the authors to carefully go through them. [1,2,3,4]


b. The reviewer also has concerns regarding the baselines chosen for comparison. In the context of HAR, there are a variety of state-of-the-art methods, based on pretraining approach to address the cross-domain challenges. Currently, it seems most baseline models are general methods such as models designed for general time series problems.

c. The experimental section might need more investigations. Correct me if I am wrong, for example, during the pretraining process, how different loss components contribution to the final performance remains unknown.


d. There are some writing issues need to be addressed. For example, in line 104, there is a refnerecne missing.

e. The reviewer is curious  about the motivation for studying the interpretability of HAR models. HAR seems like a basic tool that generates categories of activity that can be used by downstream applications. As a result, if the accuracy is good enough, it seems we do not need to understand why the model is doing good.

f. The current analysis of interpretaability might be further improved, e.g., with more examples to justify the interpretability, or why this kind of interpretability if a good fit (given there are many kinds of ways to do interpretation). Also, could you explain more about how the proposed method actually "interprets" the HAR process.


References:

[1] Cross-Dataset Activity Recognition via Adaptive Spatial-Temporal Transfer Learning, IMWUT

[2] CrossHAR: Generalizing Cross-dataset Human Activity Recognition via Hierarchical Self-Supervised Pretraining, IMWUT

[3] Generalizable Low-Resource Activity Recognition with Diverse and Discriminative Representation Learning, KDD

[4] Learning Generalizable Physiological Representations from Large-scale Wearable Data, NeurIPS

---

> ### Author Rebuttal · Authors · 2025-07-31
>
> We sincerely thank Reviewer nWUi for the thoughtful and constructive review. We strongly agree with your perspective that **generalization** is a critical challenge in wearable sensor-based human activity recognition. Our work aims to **advance the solution to this generalization problem by decomposing activities to make them more interpretable**. Below, we respond to the concerns raised, and let us know if any issues remain.
>
> ### a. Categorization and Limited Related Work
> Thank you for the valuable suggestion and for highlighting important related works. In the revised version, we will reorganize the *Related Work* section into four clearer categories to improve structure and clarity:
> - **Traditional HAR architectures and methods.** These approaches are often limited in their ability to generalize across datasets.
> - **Cross-domain and cross-dataset generalization methods.** We will expand our discussion of recent work in this area, including advanced representation learning and domain generalization techniques tailored to HAR, including the cited works [1–4].
> - **Multi-sensor composite activity recognition and compositional generalization.** This category includes methods that conceptually align with our motion primitive–based approach, particularly in their ability to handle complex and previously unseen activity compositions. We will also include relevant studies on compositional generalization from other domains, such as robotics and NLP, to provide a broader perspective.
> - **Interpretable HAR Methods.** We will compare our approach with prior work focused on interpretability in HAR (e.g., [29, 43, 44, 54] in the manuscript), highlighting both methodological differences.
>
> ### b. The reviewer also has concerns regarding the baselines chosen for comparison
> In our paper, we selected four baseline frameworks with distinct motivations:
> - **BYOL:** This choice was inspired by a summary work [r1]. **As contrastive learning is a classic framework for addressing cross-dataset challenges,** we chose a standard BYOL framework combined with three data augmentation methods shown to be effective in that work.
> - **ModCL:** This method [r2] designs specific data augmentations for the HAR task. Its authors claim that it "outperforms existing state-of-the-art (SOTA) methods" in extensive cross-dataset validation.
>   > **By extensive cross-data set validation**, experimental results show that ModCL outperforms existing  state-of-the-art (SOTA) methods on recognition accuracy metric, proving the superiority of ModCL in capturing meaningful features.
> - **TSLANet:** This work [r3] represents the rapid advancements in the general-purpose time-series domain. We included it to ensure our comparisons account for excellent research from this important upstream field.
> - **CALANet:** As a recent NeurIPS 2024 paper [r4] , we selected this model to compare our method against the latest and most advanced work specifically within the HAR domain.
>
> In response to your comment, we are actively contacting the authors of the additional method you mentioned. Meanwhile, we have added TimesNet, a strong time-series baseline, to our comparisons:
>
> ||PAMAP2|DSADS|Realworld|
> | :---: | :---: | :---: | :---: |
> |TimesNet (Acc)|86.88|97.17|92.51|
> |TimesNet (F1)|85.31|96.05|92.16|
> |Ours (Acc)|86.08|97.60|92.05|
> |Ours (F1)|84.83|97.38|93.25|
>
> [r1] What makes good contrastive learning on small-scale wearable-based tasks? SIGKDD 2022.
>
> [r2] Modality consistency-guided contrastive learning for wearable-based human activity recognition. IEEE Internet of Things.
>
> [r3] TSLANet: Rethinking Transformers for Time Series Representation Learning. ICML 2024.
>
> [r4] CALANet: Cheap All-Layer Aggregation for Human Activity Recognition. NeurIPS 2024.
>
> ### c. The experimental section might need more investigations
> - In MoPFormer, the pretraining loss combines a Masked Auto-Encoding loss ($\mathcal{L}\_{mae}$) and a Vector Quantization loss ($\mathcal{L}\_{vq}$), with an auxiliary classification loss ($\mathcal{L}\_{cls}$):
> $$
> \mathcal{L} = \lambda\_{mae}\mathcal{L}\_{mae} + \lambda\_{cls}\mathcal{L}\_{cls} + \lambda\_{vq}\mathcal{L}\_{vq}
> $$
> The two core components, $\mathcal{L}\_{mae}$ and $\mathcal{L}_{vq}$, are structurally interdependent: $\mathcal{L}\_{mae}$ predicts the quantized token indices generated by the VQ module trained with $\mathcal{L}\_{vq}$. Removing either would break the tokenization-reconstruction mechanism, making direct ablation of individual loss terms infeasible.
>
> - Instead, we indirectly assess their contribution by ablating the entire VQ module (denoted as w/o VQ), which disables both $\mathcal{L}\_{mae}$ and $\mathcal{L}\_{vq}$. The resulting performance drop confirms the tokenization mechanism is critical not only to effectiveness but also to interpretability:
>
>   |||PAMAP2|Realworld |
>   | :--: | :--: | :--: | :--: |
>   |w/o VQ| Acc|84.71|91.02|
>   ||F1| 82.42 |90.17|
>   | MoPFormer | Acc |86.08|92.05|
>   ||F1|84.83|93.25|
>
> - Additionally, Table 2 in the main text presents ablation results on key architectural components, including:
>    - w/o Stat Feature Embedding Layer
>    - w/o Metadata Embedding Adapter
>
> These experiments collectively highlight the necessity of each module and the tight integration between the loss components during pretraining.
>
> ### d. There are some writing issues need to be addressed
> Thank you again for your meticulous review. We will update it in the revised version.
>
> ### e. The reviewer is curious about the motivation for studying the interpretability of HAR models
> This is a fundamental and excellent question. We strongly agree that generalization is a core challenge for HAR. We argue that many of these challenges can be framed through the lens of compositional generalization—the ability of a model to recognize novel combinations of familiar components [r5]. To illustrate with practical challenges:
>   - Cross-Subject Generalization: Different individuals perform the same activity, like 'running', with unique styles. Our model would recognize that the core of 'running' lies in the leg motion primitives (e.g., alternating strides), while treating stylistic variations like different arm swings as separate, reusable components. This allows it to generalize to new users whose holistic running pattern has not been seen before.
>   - Cross-Dataset Generalization: Consider the activity 'ascending stairs'. In Dataset A, subjects might be instructed to climb stairs while holding the handrail. In Dataset B, subjects might perform the same activity but with their hands in their pockets. Semantically, both are 'ascending stairs', but the sensor data from the arms and torso will differ significantly. Previous methods trained only on Dataset A might fail on Dataset B because it has learned a holistic pattern that includes 'hand-on-rail' motion. In contrast, our method would learn to interpretably separate the core primitives of 'leg-lifting' and 'torso-stabilizing' from the context-specific arm primitives. This separation, when combined with prior knowledge, allows the model to correctly identify the core activity in Dataset B, even with different arm movements, demonstrating superior generalization.
>
> This highlights why interpretability is not optional—it is the key to verifying whether the model learns meaningful, reusable primitives rather than spurious correlations. For example, seeing that the model has learned a clean representation of “leg swing” gives us confidence in its ability to recombine it across contexts. In this view, building inherently interpretable models is a direct path to solving HAR’s generalization challenges [r6].
>
> [r5] Human-level concept learning through probabilistic program induction. Science.
>
> [r6] Stop explaining black box machine learning models for high stakes decisions and use interpretable models instead. Nature Machine Intelligence.
>
> ### f. The current analysis of interpretaability might be further improved
> Thank you for emphasising the need for clearer justification and explanation. Below we outline (1) why motion-primitive interpretability is well-suited to HAR and (2) how MoPFormer actually delivers it.
> - **Why motion-primitive interpretability is a good fit**
>   As argued in our response to (e), human activities are hierarchical and compositional: complex actions are built by combining simple, reusable sub-motions (e.g., a “leg-lift” appears in walking, running, and stair-climbing). Modeling these as interpretable motion primitives supports generalization across users, sensor placements, or datasets. This form of interpretability helps us understand how the model adapts to new contexts, rather than treating activities as monolithic classes.
>
> - **How MoPFormer interprets the HAR process**
>   Our model's interpretation process is conceptually analogous to how a Large Language Model (LLM) processes language. Just as sentences are composed of words, activities are composed of fine-grained motion primitives. Our model is explicitly designed to follow this logic:
>   - **Discovering motion primitives** ("build a vocabulary"): The first stage of our model uses a Vector Quantization (VQ) module with a Masked Autoencoder (MAE) objective, which is to construct a discrete codebook of effective, fundamental motion primitives from the raw, continuous sensor data.
>   - **Understanding primitive sequences** ("learn the grammar"): The subsequent Transformer module receives sequences of these discrete primitives as input. Its role is to learn the "grammar" how these primitives are combined in sequence to form complex, high-level activities.
>   - **Classification based on understanding**: Finally, the model classifies the activity based on its learned understanding of the entire primitive sequence.
>
> In the revision, we will include visualizations (e.g., token usage patterns across activities) to better illustrate how different activities rely on distinct primitive combinations, reinforcing the interpretability argument.

---

> > ### Comment · Reviewer_nWUi · 2025-08-05
> >
> > I appreciate the authors for the response, which has addressed most of my concerns, thanks! Also, thanks for the additional experimental results.
> >
> > However, I am curious about the interpretation part, how to discover motion primitives. While I like the analogy of natural language, the continuous sensor data is fundamentally different from language data. Language data is much easier to construct the vocabulary. As we know, two segments of sensor data might look significantly different even though they represent the same sub-activity.

---

> > > ### Author Response · Authors · 2025-08-06
> > >
> > > We sincerely thank the reviewer for the detailed feedback and the opportunity to further explain our work.
> > >
> > > Your follow-up question regarding the discovery of motion primitives is extremely valuable. You have pointed out the main challenge we tried to solve: **how to build a vocabulary from continuous sensor data,** which is basically different from naturally separate language data. **This was the main problem we thought about when creating our approach.**
> > >
> > > As you correctly state, in natural language processing, words give a clear, separate symbolic representation. To use a similar framework for HAR, **we must first effectively map the continuous sensor signals to a vocabulary of separate symbols** (in our case, the VQ indices). The main difficulty, which you pointed out, is that "two segments of sensor data might look significantly different even though they represent the same sub-activity."
> > >
> > > - Our solution is a multi-part effort to make sure each "vocabulary word" (VQ index) matches a consistent and simple pattern. The primary step, as you noted, depends on carefully choosing the **granularity**. We chose to define our primitives at the finest, most uniform level possible: the directional trend of a signal on a single axis from a single sensor (e.g., rising, falling, or flat).
> > >
> > > - However, even these simple trends can have different strengths (e.g., a slow arm swing vs. a fast one). To make sure these shape-similar but strength-different signals map to the same primitive, we introduce **Instance Normalization** before the VQ module. This step standardizes the input, allowing the VQ module to focus only on the pattern's shape. This choice directly addresses the problem of high within-class variation and prevents the vocabulary from becoming filled with redundant primitives for different speeds of the same motion.
> > >
> > > - Of course, normalization throws away important strength information. We solve this by bringing back this information later through the **Stat Feature Embedding Layer**. Finally, to tell whether a "rising" primitive comes from a wrist sensor or an ankle sensor, we embed the signal's source information through our **Metadata Embedding Adapter**.
> > >
> > > In summary, you are right that continuous sensor data and language data are fundamentally different. We argue that for sensor data to be understood compositionally like language, the primary task is to construct an effective vocabulary. Through a combination of fine-grained feature extraction, instance normalization, and re-embedding, we create a robust mapping from the complex continuous domain to the separate symbolic domain.

---

> > > > ### Comment · Reviewer_nWUi · 2025-08-06
> > > >
> > > > The reviewer appreciates the further explanations, thanks! Just to make sure I get the right message, you mentioned:
> > > >
> > > > "Our solution is a multi-part effort to make sure each "vocabulary word" (VQ index) matches a consistent and simple pattern. The primary step, as you noted, depends on carefully choosing the granularity. We chose to define our primitives at the finest, most uniform level possible: the directional trend of a signal on a single axis from a single sensor (e.g., rising, falling, or flat)."
> > > >
> > > > Does it mean the sensor vocabulary only consists of "rising, falling...", these directional trends for each axis? In this case, it might lose some important and detailed information. I do appreciate your discussion about how to make up for the loss of strength information, though.

---

> > > > > ### Author Response · Authors · 2025-08-06
> > > > >
> > > > > Thank you for the timely discussion and your follow-up!
> > > > >
> > > > > Because of the rules of this discussion format, I can't attach new images. However, to explore this further, we looked at the raw, normalized sequences that were grouped under the same VQ index. Earlier, I described the primitives as just “rising, falling, or flat” to keep things simple. In fact, the codebook captures a wider range of patterns. When we plot many overlapping sequences linked to the same index, we don’t just see basic trends — we also find more complex shapes, like wave-like patterns or signals that rise and then fall. Each of these forms a unique primitive. You made a very important point about the risk of losing detailed information, and this is something we’ve thought a lot about. From what I’ve seen in the learned codebook, the system tries to keep important details. For example, signals that rise or fall at different speeds (like steep vs. gradual slopes) are often separated into different indices. This helps the model keep as much of the original signal’s meaning as possible.
> > > > >
> > > > >
> > > > > Of course, turning continuous signals into symbolic tokens always leads to some information loss. But the Transformer network that follows is designed to work with this, using the token sequences to build a strong understanding based on the context. Studies like Emu3 [r1] show that this token-based approach can work very well, especially when trained on large amounts of data across different types of input.
> > > > >
> > > > > From the start of this project, we’ve been interested in how this line of work might complement or integrate with large language models. If LLMs can be equipped to understand human activity through wearable data, we believe it could open up exciting new directions for the field.
> > > > >
> > > > > [r1] Emu3: Next-Token Prediction is All You Need.

---

> > > > > > ### Comment · Reviewer_nWUi · 2025-08-07
> > > > > >
> > > > > > Thanks for the detailed answers, which are helpful. Glad to see the vocabulary is more complex than I had expected. Could you share the size of the vocabulary?

---

> > > > > > > ### Author Response · Authors · 2025-08-07
> > > > > > >
> > > > > > > Thank you for the timely discussion and your follow-up!
> > > > > > > The vocabulary size we used in the main body of the paper is 1024. We conducted an ablation study on the appropriate vocabulary size, and the details can be found in Appendix B.4, starting at line 554. In the appendix, we have also added more details regarding our experimental settings.

---

> > > > > > > > ### Comment · Reviewer_nWUi · 2025-08-08
> > > > > > > >
> > > > > > > > Dear Authors,
> > > > > > > >
> > > > > > > > Thanks for the additional explanations. Most of my concerns have been addressed properly now. I would like to raise my score and hope to see the camera-ready paper soon. Good work!

---

> > > > > > > > > ### Author Response · Authors · 2025-08-09
> > > > > > > > >
> > > > > > > > > Thank you very much for your positive feedback and support. We are delighted to hear that our explanations have addressed your concerns, and we sincerely appreciate the increased score. We look forward to preparing the camera-ready version.

---

### Official Review · Reviewer_TS74 · 2025-07-03

**Clarity:** 2
**Significance:** 3
**Originality:** 2
**Rating:** 4
**Confidence:** 3

**Summary:**

MoPFormer is a self-supervised Transformer-based framework for human activity recognition using wearable IMU sensors, designed to improve interpretability and generalisation across datasets. Borrowing from techniques successful in natural language processing,  it tokenizes raw sensor data into discrete motion codewords using vector quantization, enriched via context embedding and is trained as masked reconstruction. The proposed method is compared across datasets with SOTA baselines, and ablation studies are performed on the module components

**Questions:**

- sec 3.1: what's the motivation for extracting features per channel and not from 6-dimensional vector representing 3D motion

- Interpretability: Please provide more details and visualisations on interpretability. For example, appendix figures 5-7, while the activations are different, it seems the same motion primitives are employed in same ratios for multiple different activities. How do you ensure the codewords are used optimally?

- Most datasets used contain multiple sensors, and I believe the proposed architecture can be extended to use more than one sensor, too.

**Ethical Concerns:**

["NO or VERY MINOR ethics concerns only"]

**Final Justification:**

The authors have addressed most of my concerns. I am not entirely convinced about the interpretability of motion primitives still. I do believe this work to be high impact and the problem to be useful for the AI community

**Limitations:**

yes

**Paper Formatting Concerns:**

$\textit {Minor language errors} $
- L7: first ---> First
- Acronyms VQ, MAE, CLS appear on pages 2,3, 4 resp. but only described on pages 4, 6, 5

**Quality:**

3

**Strengths And Weaknesses:**

$\textbf {Strengths} $
- Use of codeword as motion feature embedding is novel to my knowledge.
- The authors have ablated the architecture design and parameters in the main paper and the appendix


$\textbf {Weaknesses} $
- Interpretability: The authors provide an intuitive explanation of interpretability in the introduction; however, I found the results and visualisation in the main paper and appendix insufficient to support this claim.
- While normalising each input channel is reasonable, the rationale for independently quantising each feature channel was not clearly articulated.

$\textit {Minor weaknesses} $
- Other than the type of GPU, the paper does not contain sufficient information on the experiments' compute resources
- The learnable parameter sizes in the motion primitive module and Context-Aware Embedding Module are not stated.

---

> ### Author Rebuttal · Authors · 2025-07-31
>
> Thank you for your insightful and constructive feedback. We sincerely appreciate the opportunity to clarify and improve our work. Some of your *Weaknesses* (W) and *Questions* (Q) touch upon overlapping themes, so we address them together for clarity and coherence. If you have any further comment, please feel free to let us know.
>
> ### [W1 & Q2] Interpretability
> > [W1] The authors provide an intuitive explanation of interpretability in the introduction; however, I found the results and visualisation in the main paper and appendix insufficient to support this claim.
> > [Q2] Please provide more details and visualisations on interpretability. For example, appendix figures 5-7, while the activations are different, it seems the same motion primitives are employed in same ratios for multiple different activities. How do you ensure the codewords are used optimally?
>
> Thank you for this comment. It helps us realize that our description in the paper was not sufficiently clear. We provide further justification below and will revise the manuscript to emphasize these ideas.
> - **The similarity in token usage is expected and reflects compositional structure.** Activities like 'ascending stairs', 'cycling', and 'walking' naturally share common motion elements. MoPFormer is designed to decompose complex activities into fundamental, reusable primitives. Many of these correspond to common local patterns (e.g., acceleration/deceleration along one axis). Due to instance normalization, similar patterns are mapped to the same index even if their original means or intensities differ. The model then relies on the embedded sensor metadata to distinguish whether a pattern originates from an arm accelerometer or a leg gyroscope.
> - **The sequence of tokens matters more than raw frequency.** As we demonstrate with the Markov transition matrix (line 581), the final activity classification depends critically on the sequence of primitives, not just their frequency. This is analogous to natural language, where the meaning of a sentence is determined by word order, not just word counts.
> - **The distributions in Figures 5–7 do exhibit subtle differences.** For example, Figure 7 shows a higher proportion of indices between 459-510 compared to Figures 5 and 6. Similarly, the proportion around index 765 in Figure 6 differs from the other two.
>
> ### [W2 & Q1] Per-channel quantization design
> > [W2] While normalising each input channel is reasonable, the rationale for independently quantising each feature channel was not clearly articulated.
> > [Q1] sec 3.1: what's the motivation for extracting features per channel and not from 6-dimensional vector representing 3D motion
>
> We clarify the motivation behind our design choices as follows.
> - **The core motivation is to capture fine-grained, universal primitives.** The most fundamental primitives, such as accelerating along a single axis, are often generalizable across different datasets and tasks, which promotes robustness. If we were to quantize multi-channel vectors directly, the resulting primitives would be composites of finer-grained actions. This would lead to a combinatorial explosion in the number of primitives, making it less likely that a primitive required for a target dataset was seen during pre-training.
> - **This aligns with recent findings in time-series research.** Both channel-independent (e.g., PatchTST) and channel-mixing (e.g., Autoformer [r1], FEDformer [r2]) approaches have been explored. A recent comprehensive benchmark [r3] suggests that channel-independent models tend to perform better on classification tasks. Further research [r4] indicates this phenomenon is closely related to the degree of correlation between channels.
>
> [r1] Autoformer: Decomposition Transformers with Auto-Correlation for Long-Term Series Forecasting. In NeurIPS, 2021.
>
> [r2] FEDformer: Frequency Enhanced Decomposed Transformer for Long-term Series Forecasting. In ICML, 2022.
>
> [r3] Deep Time Series Models: A Comprehensive Survey and Benchmark. arXiv preprint arXiv:2407.13278.
>
> [r4] A Closer Look at Transformers for Time Series Forecasting: Understanding Why They Work and Where They Struggle. In ICML, 2025.
>
> ### [Q3] Most datasets used contain multiple sensors, and I believe the proposed architecture can be extended to use more than one sensor, too.
> You are absolutely right that supporting multi-sensor data is essential. **Our model was explicitly designed with this** and has been evaluated on datasets that include multiple sensors. We apologize this was not sufficiently emphasized in the main text. To clarify, our architecture is inherently built to operate on multiple sensor channels, each of which is processed through motion primitive embedding layer and later integrated through metadata embeddings. The descriptions of the datasets—**all of which are multi-channel and most of which are multi-sensor**—were provided in Appendix D.
>
> ### [Minor weaknesses] Missing details on compute resources and module parameter sizes
> - **Compute Resources**: All experiments were conducted on a single NVIDIA RTX 8000 GPU. The pre-training phase took approximately 30 hours, while fine-tuning times varied by dataset, ranging from about 1 to 2.5 hours. As noted in Appendix A.1, we employed PyTorch Lightning with gradient accumulation to support a batch size of 512 and optimize memory efficiency.
> - **Model Parameter Sizes**: As described in Appendix A.1, the MoPFormer architecture includes 5 standard Transformer encoder layers, each with 8 attention heads and an MLP ratio of 1. The total number of trainable parameters is approximately 2.8 million, with the Motion Encoder contributing 2.5M. The Motion Primitive module uses a codebook size of 1024 and an embedding dimension of 256. The MAE head and Classification head contribute 263K and 1.5K parameters, respectively. Additional ablation studies related to these architectural choices are provided in Appendix B.
>
> We will ensure these details are explicitly included in the revised manuscript for completeness.
>
> ### [Minor lauguage errors]
> Thank you for pointing this out. We will correct the capitalization and ensure that all acronyms (VQ, MAE, CLS) are properly introduced upon first use.

---

> ### Comment · Reviewer_TS74 · 2025-08-07
>
> Thank you for addressing all my concerns. While I appreciate the author's effort in creating a novel work on motion primitives, the interpretability of such primitives requires more convincing evidence. The authors have addressed this in the rebuttal, and I hope that the final manuscript will include additional explanations or visualisations.
> I would retain my rating.

---

> > ### Author Response · Authors · 2025-08-09
> >
> > Thank you very much for your positive feedback and support！

---

### Official Review · Reviewer_MMnJ · 2025-07-03

**Clarity:** 3
**Significance:** 2
**Originality:** 3
**Rating:** 4
**Confidence:** 4

**Summary:**

This paper tackles interpretability and generalization issues in Human Activity Recognition (HAR) using wearable sensors by introducing MoPFormer (Motion-Primitive Transformer). MoPFormer tokenizes inertial measurement unit (IMU) signals into "motion primitive" codewords, enhancing interpretability and utilizing a Transformer to learn temporal patterns. It operates in two stages: segmenting and quantizing sensor data, followed by enriching and processing with a Transformer encoder. Pre-trained with a masked motion-modeling objective, MoPFormer outperforms state-of-the-art methods across six HAR benchmarks (PAMAP2, DSADS, MHealth, Realworld, UCI-HAR, and USC-HAD) and generalizes well to new datasets by capturing consistent movement patterns.

**Questions:**

* While it does outperform all other models, I'm noticing that MoPFormer requires pertaining to achieve these results, whereas the TSLANet achieves better performance without pre-training. Are there any learnings here?
* Tokenization: How do segment length and quantization affect performance? Is there a version of this where you could remove tokenization and run the model just on classification to understand the impact of VQ? Other ablation studies would also be valuable.
* What is the impact of noise or perturbations on the sensor inputs on tokenization? Does minor perturbation every significantly change the token distribution?

**Ethical Concerns:**

["NO or VERY MINOR ethics concerns only"]

**Final Justification:**

They were able to mediate most of my concerns with the rebuttal so I am increasing my score.

**Limitations:**

Yes.

**Quality:**

3

**Strengths And Weaknesses:**

Strengths
* The authors develop an approach to HAR using new VQ+LLM style modeling approaches. They pre-train using MAE and have a classifier head for each dataset. Overall it's a nice time-series variant of this paradigm.
* They show some ways intermediates of the models can be interpreted.
* Excels in cross-dataset performance and outperforms current methods on multiple benchmarks.

Weaknesses
* My understanding is that there are other Time-series foundation models in this direction. It would be nice to see more comparisons to these.
* Tokenization: Needs more justification on how primitives capture patterns. Are these really "motion primitives" (i.e., can you interpret them in isolation?)
* Missing Ablations: Lack of studies on component contributions weakens empirical claims.
* I appreciate seeing someone release new work using VQ+LLM and adapting it to time-series problems. It seems a little too applied in a way? This isn't necessarily problematic, but the ideas aren't especially novel.

---

> ### Author Rebuttal · Authors · 2025-07-31
>
> We sincerely thank you for the valuable comments. Below, please find our responses to each concern, and let us know if any issues remain.
> ### [Q1] While it does outperform all other models, I'm noticing that MoPFormer requires pertaining to achieve these results, whereas the TSLANet achieves better performance without pre-training. Are there any learnings here?
> - Thank you for your observation. As noted, MoPFormer is designed not only to achieve strong performance but also to introduce interpretable intermediate representations through motion primitives. These primitives enable the model to decompose complex behaviors into sequences of simple and human-understandable sub-actions, which facilitates compositional generalization to unseen activities. **This design naturally calls for a pretraining stage to learn a reusable and expressive primitive vocabulary**.
> - Regarding the comparison with TSLANet [r1], we would like to respectfully clarify that **TSLANet also involves a pretraining phase**. As stated in Appendix D of their paper:
>   > "To train the classification experiments, we optimized TSLANet using AdamW with a learning rate of 1e-3 and a weight decay of 1e-4, **applied during both training and pretraining phases.** The experiments ran for 50 epochs for **pretraining** and 100 epochs for **fine-tuning**."
>
>   As noted in our Appendix A.2, we follow the same settings. Thus, both MoPFormer and TSLANet benefit from the pretraining.
>
> [r1] TSLANet: Rethinking Transformers for Time Series Representation Learning. In ICML, 2024.
>
> ### [Q2] Tokenization: How do segment length and quantization affect performance? Is there a version of this where you could remove tokenization and run the model just on classification to understand the impact of VQ? Other ablation studies would also be valuable.
> Thank you for this insightful suggestion. We address these points as follows:
> - Segment length is indeed a critical hyperparameter. In Appendix B.1, Table 3, we conducted an ablation study by varying the **motion primitive window length**, and found that a length of 50 (i.e., 0.5s, given 100Hz resampling) provided the best results. This value was used consistently in all the experiments. We appreciate you highlighting its importance and will move this analysis into the main text in the revised version.
>
> - Following your suggestion, we conducted an additional ablation by **removing the VQ module**, effectively degrading the model into a standard Transformer Encoder (labeled "w/o VQ" in the table below). The observed performance drop across two tasks confirms that the tokenization step plays a critical role in overall model effectiveness.
>     |            |         |   PAMAP2       | Realworld  |
>     |:---:|:---:|:---:|:---:|
>     |    w/o VQ  |  Acc | 84.71  |  91.02   |
>     |      | F1  | 82.42  |   90.17   |
>     |    MoPFormer |  Acc   | 86.08  |   92.05   |
>     |      | F1  | 84.83  |   93.25   |
>
> - Additionally, we include an ablation on the **VQ codebook size** in Appendix B.4, Table 6, showing that the size of the codebook directly impacts both model performance and the expressiveness of the discovered motion primitives.
>
>
> ### [Q3] What is the impact of noise or perturbations on the sensor inputs on tokenization? Does minor perturbation every significantly change the token distribution?
> - To assess the robustness of our tokenization process, we conducted a perturbation experiment by **adding Gaussian noise to the input signals**, with variance set to 10% of the original test data variance. We then evaluated our pre-trained model under this setting (denoted by "Ours + Noise"). As shown in the table below, the performance degrades only slightly, indicating that MoPFormer is robust to moderate input perturbations and that the learned motion primitives are stable under noise.
>      |  | | PAMAP2 | DSADS | Realworld |
>      |:----:|:---:|:---:|:---:|:---:|
>      |     Ours | Acc      | 86.08  | 97.60 |   92.05   |
>      |  | F1      | 84.83  | 97.38 |   93.25   |
>      | Ours + Noise | Acc | 85.21  | 97.03 |   92.51   |
>      |  | F1  | 83.13  | 96.87 |   92.87   |
>
> - To further assess the impact of perturbations on the **token distribution**, we analyzed the VQ index sequences before and after noise addition. For a representative 'walking' sample from the PAMAP2 dataset (z-axis, wrist sensor), the VQ index sequence remained completely unchanged:
>     > **Original:** `170→772→538→170→777→538→170→772→538→339`
>     >
>     > **With Noise:** `170→772→538→170→772→538→170→772→538→339`
>
>   In contrast, a 'standing' sample, representing more stationary activity, showed slight variation in a few tokens:
>     > **Original:** `853→29→271→401→853→853→72→452→853→29`
>     >
>     > **With Noise:** `29→29→271→401→853→853→72→452→853→29`
>
>   To quantify this effect, we analyzed 500 randomly sampled windows across the dataset and found that only **2.6%** exhibited some change in their VQ token sequence after noise was added. This demonstrates that the overall token distribution remains highly stable under minor perturbations, providing a strong basis for future work on assigning human-understandable semantic labels to motion primitives.
>
> ### [W1] My understanding is that there are other Time-series foundation models in this direction. It would be nice to see more comparisons to these.
> Thank you for the suggestion. In addition to the comparisons already included in the paper (TSLANet and CALANet), we have added TimesNet [r2], a strong time-series foundation model, as an additional baseline. We used the authors' default hyperparameters and adopted a training strategy consistent with the supervised baselines (80% train, 20% test). These results demonstrate that MoPFormer remains competitive with TimesNet, even though our self-supervised methods were evaluated under a more challenging 20% train, 80% test split.
>  |  |  | PAMAP2 | DSADS | Realworld |
>  |:---:|:---:|:---:|:---:|:---:|
>  |   TimesNet | Acc    | 86.88  | 97.17 |   92.51   |
>  | | F1 | 85.31  | 96.05 |   92.16   |
>  | MoPFormer | Acc | 86.08  | 97.60 |   92.05   |
>  |  |F1 | 84.83  | 97.38 |   93.25   |
>
>  [r2] TimesNet: Temporal 2D-Variation Modeling for General Time Series Analysis. In ICLR, 2023.
>
> ### [W2] Tokenization: Needs more justification on how primitives capture patterns. Are these really "motion primitives" (i.e., can you interpret them in isolation?)
> Thank you for the insightful comment. While our current work does not assign semantic meaning to each primitive in isolation, we show that the learned tokens form distinct, repeatable patterns across similar activities (e.g., walking vs. standing), and remain stable under perturbations (the response to [Q3]).
> > **Walking**: `170→772→538→170→777→538→170→772→538→339`
> >
> > **Standing**: `853→29→271→401→853→853→72→452→853→29`
>
> We view this as an important step toward interpretability. The primitives are designed to serve as compositional units that reflect structure in the data. Assigning human-understandable labels to them—potentially via LLM-based grounding—is a possible direction for future work. We will revise the manuscript to clarify this motivation.
>
>
> ### [W3] Missing Ablations: Lack of studies on component contributions weakens empirical claims.
> Thank you for pointing this out. In addition to the new ablation on removing the VQ module (discussed in our response to [Q2]), we have already **conducted and reported several key ablation studies** in the manuscript:
> - In Table 2 of the main text, we evaluate the contributions of three core components through:
>     - w/o Pretrain Stage
>     - w/o Stat Feature Embedding Layer
>     - w/o Metadata Embedding Adapter
> - In the Appendix, we further include supplementary ablations covering:
>     - Motion Primitive Window Length (Table 3)
>     - Transformer Encoder Depth (Table 4)
>     - Hidden Embedding Dimensionality (Table 5)
>     - Vector Quantization Codebook Size (Table 6)
>
> These ablations provide a comprehensive understanding of how each module contributes to overall model performance.
>
> ### [W4] I appreciate seeing someone release new work using VQ+LLM and adapting it to time-series problems. It seems a little too applied in a way? This isn't necessarily problematic, but the ideas aren't especially novel.
> Thank you for your thoughtful feedback. While our work builds upon the VQ-based framework, we believe that its application to **decomposing complex human activities into interpretable motion primitives** represents a key and underexplored direction. Our primary innovation lies in enabling interpretable intermediate representations that move activity recognition toward a more human-aligned, compositional, and explainable paradigm. Achieving this required overcoming several domain-specific challenges:
> - To handle the wide-ranging values in raw sensor data, which would otherwise cause VQ to cluster primarily based on signal magnitude, we introduced **Instance Normalization** to standardize the inputs.
> - Since normalization removes intensity information, we reintroduced it via a **Stat Feature Embedding Layer**.
> - To generalize across datasets with inconsistent sensor configurations, we developed a **Metadata Embedding Adapter**, which explicitly encodes sensor attributes using Google's text-embedding-004 API.
>
> These components are essential for adapting the VQ+LLM pipeline to the time-series domain while maintaining interpretability. We believe that they constitute practical but impactful advances toward explainable human activity recognition.

---

> > ### Comment · Reviewer_MMnJ · 2025-08-06
> >
> > Thank you for all of the responses. These largely address my concerns and strengthen the findings in the paper.
> >
> > I am increasing my score. Nice work.

---

> > > ### Author Response · Authors · 2025-08-07
> > >
> > > Thank you for your positive feedback and recognition of our work! We're really glad our responses helped address your concerns, and we sincerely appreciate the increased score.

---

### Official Review · Reviewer_wYXz · 2025-07-04

**Clarity:** 3
**Significance:** 3
**Originality:** 3
**Rating:** 4
**Confidence:** 4

**Summary:**

MoPFormer introduces a two-stage architecture: (1) Motion Primitive Module: Tokenizes raw IMU signals into discrete codewords via VQ-based quantization; and (2) Transformer Backbone with Context-Aware Embedding: Learns temporal representations using metadata-aware embeddings and a masked autoencoding objective. The model is evaluated on six public HAR benchmarks and shows SOTA performance with compelling ablation studies and qualitative analyses.

**Questions:**

Please refer to the Weakness part.

**Ethical Concerns:**

["NO or VERY MINOR ethics concerns only"]

**Final Justification:**

It is indeed a borderline work to me. I'm happy to see this work in NeurIPS but I'm also okay if we don't have room for it this time. The authors can incorporate all the comments and feedback for substantial revision to make the work more solid.

**Limitations:**

yes

**Quality:**

3

**Strengths And Weaknesses:**

### Strengths

- The paper clearly identifies key HAR challenges (interpretability, heterogeneity) and aligns contributions tightly to them.

- Use of motion primitives learned via VQ in self-supervised HAR is a compelling idea, especially for generalization and interpretability.

- The context-aware embedding module that integrates statistical features and sensor metadata is thoughtfully designed.

- Consistent improvements over both contrastive and supervised baselines across six diverse datasets.

- Ablations and visualizations demonstrate the importance and effect of each module.

### Weaknesses

- No error bars, standard deviations, or confidence intervals are reported, weakening the empirical rigor—especially important given the small dataset sizes and minor margins in accuracy.

- While contrastive and transformer baselines are compared, the paper omits comparisons with interpretable models like decision trees or attention-based attribution techniques specifically adapted for HAR (e.g., segment attention). Are motion primitives strictly better?

- Despite the motion primitive framework, the interpretability remains primarily quantitative (e.g., frequency and similarity). There’s no human evaluation of whether the discovered primitives correspond to semantically meaningful sub-activities.

---

> ### Author Rebuttal · Authors · 2025-07-31
>
> Thank you for your constructive feedback. Below, we provide point-by-point responses to your comments. Please don’t hesitate to let us know if any concerns remain.
> ### [W1] No error bars, standard deviations, or confidence intervals are reported, weakening the empirical rigor—especially important given the small dataset sizes and minor margins in accuracy.
>
> > We thank the reviewer for highlighting this concern regarding empirical rigor. To address it, we fine-tuned our pre-trained model three additional times using different random seeds on the PAMAP2, DSADS, and RealWorld tasks. We then computed the mean and standard deviation for both accuracy and F1-score across these independent runs. The results, summarized below, demonstrate the stability of our model’s performance. And we will update these in the revised version.
> >
> >   |     | |   PAMAP2   |   DSADS    | Realworld  |
> >   |:---:|:---:|:---:|:---:|:---:|
> >   |   Original Reported |  Acc  | 86.08 |   97.60 | 92.05 |
> >   |    | F1 | 84.83 | 97.38 | 93.25 |
> >   |   New Results |  Acc | 87.49±4.10 | 97.27±0.49 | 91.32±1.07 |
> >   |    | F1 | 85.74±3.04 | 97.11±0.24 | 92.65±1.23 |
>
> ### [W2] While contrastive and transformer baselines are compared, the paper omits comparisons with interpretable models like decision trees or attention-based attribution techniques specifically adapted for HAR (e.g., segment attention). Are motion primitives strictly better?
>
> > - Thank you for this valuable suggestion. To address the concern, we included comparisons with interpretable models, e.g., **Decision Trees** and **LightGBM** [r1]. For these baselines, we  followed standard practice and extracted 13 commonly used statistical features from each sensor channel: mean, standard deviation, minimum, maximum, median, first quartile, third quartile, variance, signal energy, zero crossing rate, peak-to-peak value, skewness, and kurtosis. The results are summarized in the table below. As shown, our proposed model (MoPFormer) significantly outperforms both interpretable baselines across all three datasets in terms of both accuracy and F1-score.
> >
> >   |            |         |   PAMAP2   |   DSADS    | Realworld  |
> >   |:---:|:---:|:---:|:---:|:---:|
> >   | Decision Trees | Acc |   65.16    |   81.91    |   77.70    |
> >   | |F1  |   61.29    |   83.68    |   73.21    |
> >   |    LightGBM |Acc    |   77.11    |   87.52    |   85.64    |
> >   |  |F1     |   76.74    |   86.23    |   83.11    |
> >   |      MoPFormer|Acc      |   **87.49**    |   **97.27**    |   **91.32**    |
> >   |      |F1       |   **85.74**   |   **97.11**    |   **92.65**    |
> >
> > - While decision tree-based models provide a degree of post-hoc feature-based interpretability, they do not aim to uncover or model the compositional structure of human activities. In contrast, MoPFormer introduces an interpretable intermediate representation by explicitly decomposing activities into motion primitives. We believe that this inductive bias not only enhances the performance but also enables a more meaningful and structured path toward explainable human activity recognition.
> >
> >  [r1] LightGBM: A Highly Efficient Gradient Boosting Decision Tree. In NIPS 2017.
>
>
>
> ### [W3] Despite the motion primitive framework, the interpretability remains primarily quantitative (e.g., frequency and similarity). There’s no human evaluation of whether the discovered primitives correspond to semantically meaningful sub-activities.
>
> > - We appreciate the reviewer’s insight and agree that evaluating the semantic meaning of the learned primitives is essential. While our current analysis is quantitative, we observed that certain motion primitives consistently correspond to specific patterns, such as increasing acceleration or rotation, suggesting potential alignment with meaningful sub-actions.
> > - Due to rebuttal constraints, we cannot include new visualizations, but our findings indicate that the VQ module captures semantically coherent patterns. As noted in the manuscript, our long-term goal is to map these primitives to a human-interpretable semantic space, enabling hierarchical decomposition (e.g., "washing hands" → “turning on tap”, “applying soap”) and compositional generalization to unseen activities. This also creates opportunities to incorporate LLM priors for semantic analysis and behavioral validation. While the lack of fine-grained sub-activity annotations remains a common challenge, MoPFormer offers a solid foundation for future progress in this direction.

---

> > ### Comment · Reviewer_wYXz · 2025-08-04
> >
> > Thanks for the responses! It is indeed a borderline work to me. I'm happy to see this work in NeurIPS but I'm also okay if we don't have room for it this time. The authors can incorporate all the comments and feedback for substantial revision to make the work more solid.

---

> > > ### Author Response · Authors · 2025-08-05
> > >
> > > Thank you for your positive evaluation and constructive feedback. We are very encouraged to hear that you would be happy to see this work in NeurIPS. We will carefully incorporate all reviewers' feedback in the revised version of our paper to make the work more solid.

---

### Decision · Program_Chairs · 2025-09-17

**Decision:**

Accept (poster)

**Comment:**

The paper proposes a self-supervised framework for human activity recognition that improves interpretability and cross-dataset generalization. It tokenizes sensor signals into discrete motion primitives and processes them with a Transformer, using masked motion modeling for robust pretraining. After the rebuttal stage and duscussions between the reviewers and authors, it recieved 4 BAs. The reviewers agreed that the proposed method makes a solid contribution by introducing motion primitives for HAR, showing strong performance across six benchmarks, and providing ablations and robustness tests. The main weaknesses that remain are limited interpretability evidence, since motion primitives are only quantitatively analyzed without strong human validation, and baseline coverage gaps, with comparisons to some SOTA HAR and interpretable models still missing or under-explored. The novelty of the method was also found to be incremental based on adapting VQ+Transformer pipelines.